# PROMETHEUS: ENDOWING LOW SAMPLE AND COMMUNICATION COMPLEXITIES TO CONSTRAINED DECENTRALIZED STOCHASTIC BILEVEL LEARNING

## ABSTRACT

In recent years, constrained decentralized stochastic bilevel optimization has become increasingly important due to its versatility in modeling a wide range of multi-agent learning problems, such as multi-agent reinforcement learning and multi-agent meta-learning with safety constraints. However, one under-explored and fundamental challenge in constrained decentralized stochastic bilevel optimization is how to achieve low sample and communication complexities, which, if not addressed appropriately, could affect the long-term prospect of many emerging multi-agent learning paradigms that use decentralized bilevel optimization as a bedrock. In this paper, we investigate a class of constrained decentralized bilevel optimization problems, where multiple agents collectively solve a nonconvex-strongly-convex bilevel problem with constraints in the upper-level variables. Such problems arise naturally in many multi-agent reinforcement learning and meta learning problems. In this paper, we propose an algorithm called Prometheus (proximal tracked stochastic recursive estimator) that achieves the first $\mathcal{O}(\epsilon^{-1})$ results in both sample and communication complexities for constrained decentralized bilevel optimization, where $\epsilon > 0$ is a desired stationarity error. Collectively, the results in this work contribute to a theoretical foundation for low sample- and communication-complexity constrained decentralized bilevel learning.

## 1 INTRODUCTION

In recent years, the problem of constrained decentralized bilevel optimization has attracted increasing attention due to its foundational role in many emerging multi-agent learning paradigms with safety or regularization constraints. Such applications include, but are not limited to, safety-constrained multi-agent reinforcement learning for autonomous driving (Bennajeh et al., 2019), sparsity-regularized multi-agent meta-learning (Poon & Peyré, 2021), and rank-constrained decentralized matrix completion for recommender systems (Pochmann & Von Zuben, 2022), etc. As its name suggests, a defining feature of constrained decentralized bilevel optimization is "decentralized," which implies that the problem needs to be solved over a network without any coordination from a centralized server. As a result, all agents must rely on communications to reach a *consensus* on an optimal solution. Due to the potentially unreliable network connections and the limited computation capability at each agent, such network-consensus approaches for constrained decentralized bilevel optimization typically call for low sample and communication complexities. To date, however, none of the existing works on sample- and communication-efficient decentralized bilevel optimization in the literature considered domain constraints (e.g., Gao et al. (2022); Yang et al. (2022); Lu et al. (2022); Chen et al. (2022b) and Section 2 for detailed discussions). In light of the growing importance of constrained decentralized bilevel optimization, our goal in this paper is to fill this gap by developing sample- and communication-efficient consensus-based algorithms that can effectively handle domains constraints.

Specifically, this paper focuses on a class of constrained decentralized multi-task bilevel optimization problems, where we aim to solve a decentralized *nonconvex-strongly-convex* bilevel optimization problem with i) multiple lower-level problems and ii) consensus and domain constrains on the upper level. Such problems naturally arise in security-constrained bi-level model for integrated natural gas and electricity system (Li et al., 2017), multi-agent actor-critic reinforcement learning (Zhang et al., 2020) and constraint meta-learning (Liu et al., 2019). In the optimization literature, a natural

approach for handling domain constraints is the *proximal operator*. However, as will be shown later, proximal algorithm design and theoretical analysis for constrained decentralized bilevel optimization problems is much more complicated than those of unconstrained counterparts and the results are very limited. In fact, in the literature, the proximal operator for constrained bilevel optimization has been under-explored even in the single-agent setting, not to mention the more complex multi-agent settings. The most related works in terms of handling domain constraints can be found in (Hong et al., 2020; Chen et al., 2022a; Ghadimi & Wang, 2018), which rely on direct projected (stochastic) gradient descent to solve the constrained bilevel problem. In contrast, our work considers general domain constraints that require evaluation of proximal operators in each iteration. Also, these works only considered the single-agent setting, and hence their techniques are not implementable over networks. Actually, up until this work, it is unclear how to design proximal algorithms to handle domain constraints for decentralized bilevel optimization. Moreover, it is worth noting that existing methods for hyper-gradient approximation in both single- and multi-agent bilevel optimization are either based on first-order Taylor-type approximation approaches (Khanduri et al., 2021; Ghadimi & Wang, 2018; Hong et al., 2020), implicit differentiation (Ghadimi & Wang, 2018; Gould et al., 2016; Ji et al., 2021), or iterative differentiation (Franceschi et al., 2017; Maclaurin et al., 2015; Ji et al., 2021), all of which suffer from high communication and sample complexities that are problematic in decentralized settings over networks.

The main contribution of this paper is that we propose a series of new *proximal-type* algorithmic techniques to overcome the challenges mentioned above and achieve low sample and communication complexities for constrained decentralized bilevel optimization problem. The main technical contributions of this work are summarized below:

- We propose a decentralized optimization approach called Prometheus (proximal tracked stochastic recursive estimator), which is a cleverly designed hybrid algorithm that integrates proximal operations, recursive variance reduction, lower-level gradient tracking, and upper-level consensus techniques. We show that, to achieve an $\epsilon$-stationary point, Prometheus enjoys a convergence rate of $\mathcal{O}(1/T)$, where $T$ is the maximum number of iterations. This implies $\mathcal{O}(\epsilon^{-1})$ communication complexity and $\mathcal{O}(\sqrt{n}K\epsilon^{-1} + n)$ sample complexity per agent.

- We propose a new hyper-gradient estimator for the upper-level function, which leads to a far more accurate stochastic estimation than the conventional stochastic estimator used in (Khanduri et al., 2021; Ghadimi & Wang, 2018; Hong et al., 2020; Liu et al., 2022). We show that our new hyper-gradient stochastic estimator has a smaller variance and outperforms existing estimators both theoretically and experimentally. We note that our proposed estimator could be of independent interest for other bilevel optimization problems.

- We reveal an interesting insight that the variance reduction in Prometheus is not only sufficient but also necessary in the following sense: a "non-variance-reduced" special version of Prometheus could only achieve a much slower $\mathcal{O}(1/\sqrt{T})$ convergence to a constant error-ball rather than an $\epsilon$-stationary point with arbitrarily small $\epsilon$-tolerance. This insight advances our understanding and state of the art of algorithm design for constrained decentralized bilevel optimization.

The rest of the paper is organized as follows. In Section 2, we review related literature. In Section 3, we provide the preliminaries of the decentralized bilevel optimization problem. In Section 4, we provide details on our proposed Prometheus algorithm, including the convergence rate, communication complexity, and sample complexity results. Section 5 provides numerical results to verify our theoretical findings and Section 6 concludes this paper.

## 2 RELATED WORK

In this section, we first provide a quick overview of the state-of-the-art on single-agent constrained bilevel optimization as well as decentralized bilevel optimization.

**1) Constrained Bilevel Optimization in the Single-Agent Setting:** As mentioned in Section 1, various techniques have been proposed to solve single-agent bilevel optimization, such as utilizing full-gradient-based techniques (e.g., AID-based methods (Rajeswaran et al., 2019; Franceschi et al., 2018; Ji et al., 2021), ITD-based methods (Pedregosa, 2016; Maclaurin et al., 2015; Ji et al., 2021)), stochastic gradient-based techniques (Ghadimi & Wang, 2018; Khanduri et al., 2021; Guo & Yang, 2021), STORM-based techniques (Cutkosky & Orabona, 2019), and VR-based techniques (Yang

et al., 2021). However, none of these existing works have considered domain constraints. To our knowledge, the only works that considered domain constraints in the single-agent setting can be found in (Hong et al., 2020; Chen et al., 2022a; Ghadimi & Wang, 2018). In (Ghadimi & Wang, 2018), the authors proposed a double-loop algorithm called BSA, where in the inner loop the lower level problem is solved to sufficient accuracy, while in the outer loop projected (stochastic) gradient descent is utilized to update the model parameters. The double-loop structure of BSA led to slow convergence. In (Hong et al., 2020), a two-timescale single loop stochastic approximation (TTSA) algorithm based on projected (stochastic) gradient descent was proposed to solve the constrained bilevel optimization problems. However, TTSA has to choose step-sizes of different orders for the upper and lower level problems to ensure convergence, which leads to suboptimal complexity results. Later in (Chen et al., 2022a), an algorithm called STABLE algorithm is proposed to utilize a momentum-based gradient estimator and combines the Moreau-envelop-based analysis to achieve an $\mathcal{O}(\epsilon^{-2})$ sample-complexity. As mentioned in Section 1, however, the methods in (Ghadimi & Wang, 2018; Hong et al., 2020; Chen et al., 2022a) consider only simple constraints. Moreover, the aforementioned methods are not applicable in the decentralized setting.

**2) Decentralized Bilevel Optimization:** Decentralized bilevel optimization has also received increasing attention in recent years. For example, Yang et al. (2022), Lu et al. (2022) and Chen et al. (2022b) respectively proposed stochastic gradient (SG)-type decentralized algorithms for bilevel optimization and achieve an $\mathcal{O}(\epsilon^{-2})$ sample-communication complexity. The VRDBO method in (Gao et al., 2022) employed the momentum-based techniques for decentralized bilevel optimization to achieve better $\mathcal{O}(\epsilon^{-1.5})$ complexity results. However, VRDBO updates upper- and lower-level variables in an alternating fashion. As will be shown later, our Prometheus algorithm updates upper-level and lower-level variables *simultaneously*, which renders a much lower implementation complexity than VRDBO. Besides, Prometheus achieves $\mathcal{O}(\sqrt{n}K\epsilon^{-1} + n)$ sample complexities, which is a near-optimal sample complexity and outperforms existing decentralized bilevel algorithms. It is worth noting that, the in aforementioned works, consensus requirements exist on both lower- and upper-level subproblems. To certain extent, such a formulation can be viewed as multiple agents collaboratively solving the same bilevel optimization problem. In contrast, our work only has a consensus requirement in the upper-level subproblem, which implies multiple different lower-level tasks. This is more practically-relevant and a more appropriate formulation for multi-agent reinforcement learning, multi-agent meta-learning, etc. We note that the most related work on decentralized bilevel optimization is (Liu et al., 2022), which also considered multiple lower-level tasks. However, the INTERACT method in (Liu et al., 2022) is *unconstrained* and *cannot* handle *non-smooth* objectives considered in our work. In contrast, we propose a special proximal operator $\tilde{\mathbf{x}}_{i,t}$ to address this challenge. Last but not least, we note that *none* of the aforementioned works on decentralized bilevel optimization took domain constraints into consideration. For clearer comparisons, we summarize and compare the complexity results of all algorithms mentioned above in Table 1.

## 3 PROBLEM FORMULATION AND MOTIVATING APPLICATIONS

**1) Network Consensus Formulation for Decentralized Bilevel Optimization:** Consider an undirected connected network $\mathcal{G} = (\mathcal{N}, \mathcal{L})$ that represents a peer-to-peer network, where $\mathcal{N}$ and $\mathcal{L}$ are the sets of agents (nodes) and edges, respectively, with $|\mathcal{N}| = m$. Each agent $i$ has local computation capability and can share information with its neighboring agents denoted as $\mathcal{N}_i \triangleq \{i' \in \mathcal{N} : (i, i') \in \mathcal{L}\}$. Each agent $i$ has access to a local dataset of size $n$. All agents in the network collaboratively solve the following constrained decentralized bilevel optimization problem:

$$\min_{\mathbf{x}_i \in \mathcal{X}} \frac{1}{m} \sum_{i=1}^{m} [\ell(\mathbf{x}_i) + h(\mathbf{x}_i)] \triangleq \frac{1}{mn} \sum_{i=1}^{m} \sum_{j=1}^{n} [f\left(\mathbf{x}_i, \mathbf{y}_i^*(\mathbf{x}_i; \bar{\xi}_{ij})\right) + h(\mathbf{x}_i)]$$

$$\text{s.t. } \mathbf{y}_i^*(\mathbf{x}_i) = \arg\min_{\mathbf{y}_i \in \mathbb{R}^{p_2}} g(\mathbf{x}_i, \mathbf{y}_i) \triangleq \frac{1}{n} \sum_{j=1}^{n} g(\mathbf{x}_i, \mathbf{y}_i; \zeta_{ij}), \ \forall i; \quad \mathbf{x}_i = \mathbf{x}_{i'}, \ \text{if} \ (i, i') \in \mathcal{L}, \quad (1)$$

where $\mathcal{X} \subseteq \mathbb{R}^{p_1}$ is a convex constraint set, and $\mathbf{x}_i \in \mathcal{X}$ and $\mathbf{y}_i \in \mathbb{R}^{p_2}$ are parameters to be trained for the upper-level and lower-level subproblems at agent $i$, respectively. Here, $\ell(\mathbf{x}_i) \triangleq f(\mathbf{x}_i, \mathbf{y}_i^*(\mathbf{x}_i)) = \frac{1}{n} \sum_{j=1}^{n} f\left(\mathbf{x}_i, \mathbf{y}_i^*(\mathbf{x}_i); \bar{\xi}_{ij}\right)$ is the local objective function, and $h(\mathbf{x}_i)$ is a convex proximal function (possibly non-differentiable) for regularization. The equality constraints $\mathbf{x}_i = \mathbf{x}_{i'}$ ensure that the local copies at connected agents $i$ and $i'$ are equal to each other, hence the name "consensus form."

Table 1: Comparisons among algorithms for bilevel optimization problems. Sample complexities (both upper and lower) as defined in the sense of achieving an $\epsilon$-stationary point defined in (2), $n$ is the size of dataset at each agent.

| Algorithms | Constriants | Samp. Complex. | Comm. Complex. | Decentralized |
|---|---|---|---|---|
| BSA (Ghadimi & Wang, 2018) | ✓ | $\mathcal{O}(\epsilon^{-2})$ | - | ✗ |
| SUSTAIN (Khanduri et al., 2021) | ✗ | $\tilde{\mathcal{O}}(\epsilon^{-1.5})$ | - | ✗ |
| RSVRB (Guo & Yang, 2021) | ✗ | $\mathcal{O}(\epsilon^{-1.5})$ | - | ✗ |
| VRBO (Yang et al., 2021) | ✗ | $\mathcal{O}(\epsilon^{-1.5})$ | - | ✗ |
| AID-BiO /ITD-BiO Ji et al. (2021) | ✗ | $\mathcal{O}(n\epsilon^{-1})$ | - | ✗ |
| TTSA (Hong et al., 2020) | ✓ | $\mathcal{O}(\epsilon^{-5/2})$ | - | ✗ |
| STABLE (Chen et al., 2022a) | ✓ | $\mathcal{O}(\epsilon^{-2})$ | - | ✗ |
| DSBO (Yang et al., 2022) | ✗ | $\mathcal{O}(\epsilon^{-2})$ | $\mathcal{O}(\epsilon^{-2})$ | ✓ |
| SPDB (Lu et al., 2022) | ✗ | $\mathcal{O}(\epsilon^{-2})$ | $\mathcal{O}(\epsilon^{-2})$ | ✓ |
| DSBO (Chen et al., 2022b) | ✗ | $\mathcal{O}(\epsilon^{-2})$ | $\mathcal{O}(\epsilon^{-2})$ | ✓ |
| VRDBO (Gao et al., 2022) | ✗ | $\mathcal{O}(\epsilon^{-1.5})$ | $\mathcal{O}(\epsilon^{-1.5})$ | ✓ |
| INTERACT (Liu et al., 2022) | ✗ | $\mathcal{O}(n\epsilon^{-1})$ | $\mathcal{O}(\epsilon^{-1})$ | ✓ |
| INTERACT-VR Liu et al. (2022) | ✗ | $\mathcal{O}(\sqrt{n}K\epsilon^{-1}+n)$ | $\mathcal{O}(\epsilon^{-1})$ | ✓ |
| **Prometheus [Ours.]** | ✓ | $\mathcal{O}(\sqrt{n}K\epsilon^{-1}+n)$ | $\mathcal{O}(\epsilon^{-1})$ | ✓ |

Next, we define the notion of $\epsilon$-stationarity point for Problem (1) for convergence performance characterization. We say that $\{\mathbf{x}_i, \mathbf{y}_i, \forall i \in [m]\}$ is an $\epsilon$-stationarity point if it satisfies:

$$\underbrace{\mathbb{E}\|\tilde{\mathbf{x}} - \mathbf{1}\otimes\bar{\mathbf{x}}\|^2}_{\text{Saddle point error}} + \underbrace{\mathbb{E}\|\mathbf{x} - \mathbf{1}\otimes\bar{\mathbf{x}}\|^2}_{\text{Consensus error}} + \underbrace{\mathbb{E}\|\mathbf{y} - \mathbf{y}^*\|^2}_{\text{lower problem error}} \leq \epsilon, \qquad (2)$$

where $\bar{\mathbf{x}} \triangleq \frac{1}{m}\sum_{i=1}^m \mathbf{x}_i$, $\mathbf{y} \triangleq [\mathbf{y}_1^\top,...\mathbf{y}_m^\top]^\top$, and $\mathbf{y}^* \triangleq [\mathbf{y}_1^{*\top},...\mathbf{y}_m^{*\top}]^\top$, and $\tilde{\mathbf{x}}$ is a proximal point that will be defined later in Section 4. The first term in (2) quantifies the convergence of the $\bar{\mathbf{x}}$ to a proximal point of stationarity of the global objective. The second term in (2) measures the consensus error among local copies of the upper variable, while the last term in (2) quantifies the (aggregated) error in the lower problem's iterates across all agents. Thus, $\epsilon \to 0$ implies that the algorithm achieves three goals simultaneously: i) consensus of upper variables, ii) stationary point of Problem (1), and iii) solution to the lower problem. As mentioned in Section 1, two of the most important performance metrics in decentralized optimization are the sample and communication complexities.

**2) Motivating Applications:** Problem (1) arises naturally from many interesting real-world applications. Here, we present two motivating applications to showcase its practical relevance:

- *Multi-agent meta-learning (Rajeswaran et al., 2019):* Meta-learning (or learning to learn) is to find model that can adapt to multiple related tasks. A popular meta-learning framework is the model-agnostic meta learning (MAML), which minimizes an upper objective of empirical risk on all tasks. Consider a multi-agent meta-learning task with $m$ lower level problems and $m$ agents collectively solve this meta-learning problem over a network. This problem can be formulated as:

$$\min_{\mathbf{x}\in\mathcal{X}} \sum_{i=1}^m f\left(\mathbf{x}, \mathbf{y}_i^*(\mathbf{x})\right), \text{s.t.} \ \mathbf{y}_i^*(\mathbf{x}) \in \operatorname*{argmin}_{\mathbf{y}_i\in\mathbb{R}^{p_2}} g\left(\mathbf{x}, \mathbf{y}_i\right), i=1,\ldots,m. \qquad (3)$$

Here, agent $i$ has a local dataset with $n$ samples, $\mathbf{x} \in \mathcal{X}$ is the constrained (e.g., due to safety) model parameters shared by all agents, and $\mathbf{y}_i$ are task-specific parameters solved by each agent.

- *Decentralized min-max optimization (Huang et al., 2022):* Another application of the constrained decentralized bilevel optimization in (1) is the decentralized nonconvex strongly-concave min-max optimization problem, which is typically seen in, e.g., multi-agent reinforcement learning (Zhang et al., 2021), fair multi-agent machine learning (Baharlouei et al., 2019), and data poisoning attack (Liu et al., 2020b). A decentralized min-max optimization problem is a special case of a decentralized bilevel optimization problem because:

$$\min_{\mathbf{x}\in\mathcal{X}} \max_{\substack{\mathbf{y}_i\in\mathbb{R}^{p_2}\\i=1,\ldots,m}} \sum_{i=1}^m f\left(\mathbf{x}, \mathbf{y}_i\right) \Longleftrightarrow \min_{\mathbf{x}\in\mathcal{X}} \sum_{i=1}^m f\left(\mathbf{x}, \mathbf{y}_i\left(\mathbf{x}^*\right)\right), \text{s.t.} \ \mathbf{y}_i^*(\mathbf{x}) = \operatorname*{argmin}_{\mathbf{y}_i\in\mathbb{R}^{p_2}} -f\left(\mathbf{x}, \mathbf{y}_i\right), \forall i.$$

# 4 SOLUTION APPROACH

In this section, we first present the Prometheus algorithm for solving the constrained decentralized bilevel optimization problems in Problem (1) in Sections 4.1–4.2. Then, we provide its theoretical convergence guarantees in Section 4.3. Lastly, we will reveal a key insight on the necessity of using the proposed variance reduction techniques in Section 4.4. Due to space limitation, we relegate the proofs and the notation Table. 2 to supplementary material.

## 4.1 PRELIMINARIES

To present the Prometheus algorithm, we first introduce several basic components as preparation.

**1) Network-Consensus Matrix:** Our Prometheus algorithm is based on the network-consensus mixing approach: in each iteration, every agent exchanges and aggregates neighboring information through a consensus weight matrix $\mathbf{M} \in \mathbb{R}^{m \times m}$. We define $\lambda$ as the second largest eigenvalue of the matrix $\mathbf{M}$. Let $[\mathbf{M}]_{ii'}$ represent the element in the $i$-th row and the $i'$-th column in $\mathbf{M}$. The choice of $\mathbf{M}$ should satisfy the following properties: (a) *doubly stochastic:* $\sum_{i=1}^{m}[\mathbf{M}]_{ii'} = \sum_{j=1}^{m}[\mathbf{M}]_{ii'} = 1$; (b) *symmetric:* $[\mathbf{M}]_{ii'} = [\mathbf{M}]_{i'i}, \forall i, i' \in \mathcal{N}$; and (c) *network-defined sparsity:* $[\mathbf{M}]_{ii'} > 0$ if $(i, i') \in \mathcal{L}$; otherwise $[\mathbf{M}]_{ii'} = 0, \forall i, i' \in \mathcal{N}$.

**2) Stochastic Estimators:** In Prometheus, we need to estimate the stochastic gradient of the bilevel problem using the implicit function theorem. We note that in the literature of bilevel optimization with stochastic gradient, a commonly adopted stochastic gradient estimator is of the form (Khanduri et al., 2021; Ghadimi & Wang, 2018; Hong et al., 2020; Liu et al., 2022):

$$\bar{\nabla} f(\mathbf{x}_{i,t}, \mathbf{y}_{i,t}; \bar{\xi}_{ij}) = \nabla_{\mathbf{x}} f(\mathbf{x}_{i,t}, \mathbf{y}_{i,t}; \xi_i^0) - \frac{1}{L_g} \nabla_{\mathbf{xy}}^2 g(\mathbf{x}_{i,t}, \mathbf{y}_{i,t}; \zeta_i^0) \hat{\mathbf{H}}_{i,k} \nabla_{\mathbf{y}} f(\mathbf{x}_{i,t}, \mathbf{y}_{i,t}; \xi_i^0), \quad (4)$$

where $\hat{\mathbf{H}}_{i,k} \triangleq K \prod_{p=1}^{k(K)}(\mathbf{I} - \frac{\nabla_{\mathbf{yy}}^2 g(\mathbf{x}_{i,t}, \mathbf{y}_{i,t}; \zeta_i^p)}{L_g})$. Here, $K \in \mathbb{N}$ is a predefined parameter and $k(K) \sim \mathcal{U}\{0, \ldots, K-1\}$ is an integer-valued random variable uniformly chosen from $\{0, \ldots, K-1\}$. It can be shown that $\hat{\mathbf{H}}_{i,k}$ is a biased estimator for the Hessian inverse $[\nabla_{\mathbf{yy}}^2 g(\mathbf{x}, \mathbf{y}; \zeta)]^{-1} = \sum_{i=1}^{\infty}(\mathbf{I} - \nabla_{\mathbf{yy}}^2 g(\mathbf{x}, \mathbf{y}; \zeta))^i$. However, this estimator has the limitation that it only incorporates the *first* term in the Taylor approximation, thus resulting in a large variance and could eventually increase the communication complexity of decentralized bilevel optimizaiton.

To address this issue, in this paper, we propose a *new stochastic gradient estimator* as follows:

$$\mathbf{H}_{i,0} = \mathbf{I}; \mathbf{H}_{i,k} = \mathbf{I} + \left(\mathbf{I} - \frac{\nabla_{\mathbf{yy}}^2 g(\mathbf{x}_{i,t}, \mathbf{y}_{i,t}; \zeta_i^k)}{L_g}\right) \mathbf{H}_{i,k-1} = \mathbf{I} + \sum_{j'=1}^{k(K)} \prod_{p=1}^{j'} \left(\mathbf{I} - \frac{\nabla_{\mathbf{yy}}^2 g(\mathbf{x}_{i,t}, \mathbf{y}_{i,t}; \zeta_i^p)}{L_g}\right);$$

$$\bar{\nabla} f(\mathbf{x}_{i,t}, \mathbf{y}_{i,t}; \bar{\xi}_{ij}) = \nabla_{\mathbf{x}} f(\mathbf{x}_{i,t}, \mathbf{y}_{i,t}; \xi_i^0) - \frac{1}{L_g} \nabla_{\mathbf{xy}}^2 g(\mathbf{x}_{i,t}, \mathbf{y}_{i,t}; \zeta_i^0) \mathbf{H}_{i,k} \nabla_{\mathbf{y}} f(\mathbf{x}_{i,t}, \mathbf{y}_{i,t}; \xi_i^0). \quad (5)$$

Compared to the conventional estimator, the key difference in our new estimator lies in the matrix $\mathbf{H}_{i,k}$. The new Hessian inverse estimator is inspired by ideas in stochastic second-order optimization (Agarwal et al., 2016). Similar technique to estimate the Hessian inverse can also be found in Koh & Liang (2017). However, our Hessian inverse estimator differ from Koh & Liang (2017) in the following key aspects: (i) In our Hessian inverse estimator, we multiply the hessian term $\nabla_{yy}^2 g(x, y; \xi)$ by $1/L_g$ as it ensures that $1/L_g \times \nabla_{yy}^2 g(x, y; \xi)$ will have eigenvalue less than 1. Otherwise, the power series of the Hessian Inverse will not converge. In comparison, Koh & Liang (2017) does not have $1/L_g$ term because the authors assume w.l.o.g. that the Hessian $\nabla_{yy}^2 g(x, y) \preceq 1$, which implies that the authors implicitly assume $L_g = 1$. (ii) Koh & Liang (2017) is only designed for solving a conventional single-level minimization problem with loss function $L(\cdot)$. In comparision, our proposed stochastic estimator can be used in bilevel learning especially for solving non-smooth regularizers in the upper-level problems. Note that our $\mathbf{H}_{i,k}$ is in a recursive form that is able to capture the *entire* Taylor series at once without increasing the sample complexity. Thanks to this recursive form, $\mathbf{H}_{i,k}$ utilizes $O(k^2)$ samples, as opposed to only $O(k)$ samples in the conventional $\hat{\mathbf{H}}_{i,k}$-Hessian inverse estimator, thus leading to a much smaller variance and eventually much lower communication complexity. It is worth noting that although our $\mathbf{H}_{i,k}$ estimator leverages more training samples, the computation cost is the *same* as that of $\hat{\mathbf{H}}_{i,k}$ due to the recursive structure in (5). In Sections 4.3 and 5, we will theoretically and numerically demonstrate the smaller variance of our new estimator over the conventional one.

### 4.2 THE Prometheus ALGORITHM.

Our Prometheus algorithm is an advanced triple-hybrid of proximal, gradient tracking, and variance reduction techniques. The procedure of Prometheus can be organized into three key steps:

- *Step 1 (Local Proximal Operations):* In each iteration $t$, each agent $i$ performs the following proximal operations to cope with the domain constraint set $\mathcal{X}$ for the upper-level variables:

$$\widetilde{\mathbf{x}}_{i,t} = \tilde{\mathbf{x}}_i(\mathbf{x}_{i,t}) = \arg\min_{\mathbf{x}\in\mathcal{X}}[\langle\mathbf{u}_{i,t}, \mathbf{x} - \mathbf{x}_{i,t}\rangle + \frac{\tau}{2}\|\mathbf{x} - \mathbf{x}_{i,t}\|^2 + h(\mathbf{x})], \qquad (6)$$

where $\tau > 0$ is a proximal control parameter and $\mathbf{u}_{i,t}$ is an auxiliary vector. The proximal update rule is motivated by the SONATA method (Scutari & Sun, 2019) used in a decentralized minimization.

- *Step 2 (Consensus Update in Upper-Level Variables):* Next, each agent $i$ updates the upper and lower model parameters $\mathbf{x}_i, \mathbf{y}_i$ as follows:

$$\mathbf{x}_{i,t+1} = \sum_{i'\in\mathcal{N}_i}[\mathbf{M}]_{ii'}\mathbf{x}_{i',t} + \alpha(\tilde{\mathbf{x}}_i(\mathbf{x}_{i,t}) - \mathbf{x}_{i,t}), \qquad \mathbf{y}_{i,t+1} = \mathbf{y}_{i,t} - \beta\mathbf{v}_{i,t}, \qquad (7)$$

where $\alpha$ and $\beta$ are constant step-sizes for updating $\mathbf{x}$- and $\mathbf{y}$-variables, respectively. Note that updating $\mathbf{x}_{i,t+1}$ in Eq. (7) is a local weighted average at agent $i$ and plus a local update in the spirit of Frank-Wolfe given a proximal point. The right-hand side of Eq. (7) performs a local stochastic gradient descent update for the $\mathbf{y}$-variable at each agent $i$.

*Remark* 1. The used auxiliary proximal operator $\tilde{\mathbf{x}}_{i,t}$ and the resultant local update $\alpha(\tilde{\mathbf{x}}_i(\mathbf{x}_{i,t}) - \mathbf{x}_{i,t})$ in the consensus step play an important role in helping us alleviate the non-smooth objective challenge. It will be difficult to achieve convergence guarantees in decentralized learning if we use $\mathbf{x}_{i,t+1} = \mathcal{P}_{\mathcal{X}}(\mathbf{x}_{i,t} - \alpha\mathbf{u}_{i,t}) = \arg\min_{\tilde{\mathbf{x}}\in\mathcal{X}}\|\tilde{\mathbf{x}} - (\mathbf{x}_{i,t} - \alpha\mathbf{u}_{i,t})\|^2$ instead. See proof details in Lemma 5 and 7 in our Appendix.

- *Step 3 (Local Variance-Reduced Stochastic Gradient Estimate):* In the local gradient estimator step, each agent $i$ estimates its local gradients using the following stochastic gradient estimators:

$$\mathbf{p}_i(\mathbf{x}_{i,t}, \mathbf{y}_{i,t}) = \begin{cases} \bar{\nabla}f(\mathbf{x}_{i,t}, \mathbf{y}_{i,t}) = \frac{1}{n}\sum_{j=1}^n\bar{\nabla}f(\mathbf{x}_{i,t}, \mathbf{y}_{i,t}; \bar{\xi}_{ij}), & \text{if } \mathrm{mod}(t,q) = 0, \\ \mathbf{p}_i(\mathbf{x}_{i,t-1}, \mathbf{y}_{i,t-1}) \\ + \frac{1}{|\mathcal{S}_{i,t}|}\sum_{j\in\mathcal{S}_{i,t}}\left(\bar{\nabla}f\left(\mathbf{x}_{i,t}, \mathbf{y}_{i,t}, \bar{\xi}_{ij}\right) - \bar{\nabla}f\left(\mathbf{x}_{i,t-1}, \mathbf{y}_{i,t-1}, \bar{\xi}_{ij}\right)\right), \end{cases} \qquad (8a)$$

$$\mathbf{d}_i(\mathbf{x}_{i,t}, \mathbf{y}_{i,t}) = \begin{cases} \bar{\nabla}g(\mathbf{x}_{i,t}, \mathbf{y}_{i,t}) = \frac{1}{n}\sum_{i=1}^n\bar{\nabla}g(\mathbf{x}_{i,t}, \mathbf{y}_{i,t}; \zeta_{ij}), & \text{if } \mathrm{mod}(t,q) = 0, \\ \mathbf{d}_i(\mathbf{x}_{i,t-1}, \mathbf{y}_{i,t-1}) \\ + \frac{1}{|\mathcal{S}_{i,t}|}\sum_{j\in\mathcal{S}_{i,t}}\left(\nabla g\left(\mathbf{x}_{i,t}, \mathbf{y}_{i,t}, \zeta_{ij}\right) - \nabla g\left(\mathbf{x}_{i,t-1}, \mathbf{y}_{i,t-1}, \zeta_{ij}\right)\right). \end{cases} \qquad (8b)$$

Here, $\mathcal{S}_{i,t}$ is the sample mini-batch in the $t$-th iteration, and $q$ is a pre-determined inner loop iteration number. The local stochastic gradient estimation is a recursive estimator that shares some structural similarity with those in SARAH (Nguyen et al., 2017), SPIDER (Fang et al., 2018), and PAGE (Li et al., 2021) used for traditional minimization problems.

- *Step 4 (Gradient Tracking in Upper-Level Parameters):* Each agent $i$ updates $\mathbf{u}_{i,t}$ and $\mathbf{v}_{i,t}$ by averaging over its neighboring tracked gradients:

$$\mathbf{u}_{i,t} = \sum_{i'\in\mathcal{N}_i}[\mathbf{M}]_{ii'}\mathbf{u}_{i',t-1} + \mathbf{p}_i(\mathbf{x}_{i,t}, \mathbf{y}_{i,t}) - \mathbf{p}_i(\mathbf{x}_{i,t-1}, \mathbf{y}_{i,t-1}); \qquad \mathbf{v}_{i,t} = \mathbf{d}_i(\mathbf{x}_{i,t}, \mathbf{y}_{i,t}). \qquad (9)$$

To summarize, we illustrate the Prometheus algorithm in Algorithm 1.

### 4.3 CONVERGENCE PERFORMANCE ANALYSIS OF THE Prometheus ALGORITHM

Now, we focus on the convergence performance analysis for the proposed Prometheus algorithm. Before presenting the main convergence results, we first state several technical assumptions:

*Assumption* 1. For all $\zeta \in \mathrm{supp}(\pi_g)$ where $\mathrm{supp}(\pi)$ is the support of $\pi$, $\mathbf{x}\in\mathcal{X}$, $\mathcal{X}\subseteq\mathbb{R}^{p_1}, \mathbf{y}\in\mathbb{R}^{p_2}$, the lower-level function $g$ has the following properties : i) $g(\mathbf{x}, \mathbf{y}; \zeta)$ is $\mu_g$-strongly convex with $\mu_g > 0$, $\nabla_{\mathbf{y}}g(\mathbf{x}, \mathbf{y}; \zeta)$ is $L_g$-Lipschitz continuous with $L_g > 0$; ii) $\left\|\nabla^2_{\mathbf{xy}}g(\mathbf{x}, \mathbf{y}; \zeta)\right\|^2 \le C_{g_{xy}}$ for some $C_{g_{xy}} > 0$, $\nabla^2_{\mathbf{xy}}g(\mathbf{x}, \mathbf{y}; \zeta)$ and $\nabla^2_{\mathbf{yy}}g(\mathbf{x}, \mathbf{y}; \zeta)$ are Lipschitz continuous with constants $L_{g_{xy}} > 0$ and $L_{g_{yy}} > 0$, respectively.

---

**Algorithm 1** The Prometheus Algorithm at Each Agent $i$.

---

Set parameter pair $(\mathbf{x}_{i,0}, \mathbf{y}_{i,0}) = (\mathbf{x}^0, \mathbf{y}^0)$.
Calculate local gradients: $\mathbf{u}_{i,0} = \bar{\nabla} f(\mathbf{x}_{i,0}, \mathbf{y}_{i,0}); \mathbf{v}_{i,0} = \nabla_{\mathbf{y}} g(\mathbf{x}_{i,0}, \mathbf{y}_{i,0})$;
**for** $t = 1, \cdots, T$ **do**
    Update local parameters $(\mathbf{x}_{i,t+1}, \mathbf{y}_{i,t+1})$ as in Eq. (6)-(7);
    **if** Prometheus : **then**
        Compute local estimators $(\mathbf{p}_i(\mathbf{x}_{i,t+1}, \mathbf{y}_{i,t+1}), \mathbf{d}_i(\mathbf{x}_{i,t+1}, \mathbf{y}_{i,t+1}))$ as in Eq. (8);
    **end if**
    **if** Prometheus-SG: **then**
        Compute local estimators $(\mathbf{p}_i(\mathbf{x}_{i,t+1}, \mathbf{y}_{i,t+1}), \mathbf{d}_i(\mathbf{x}_{i,t+1}, \mathbf{y}_{i,t+1}))$ as in Eq. (10);
    **end if**
    Track global gradients $(\mathbf{u}_{i,t+1}, \mathbf{v}_{i,t+1})$ as in Eq. (9);
**end for**

---

*Assumption* 2. For all $\xi \in \operatorname{supp}(\pi_f)$ where $\operatorname{supp}(\pi)$ is the support of $\pi$, $\mathbf{x} \in \mathcal{X}, \mathcal{X} \subseteq \mathbb{R}^{p_1}$, the upper-level function $f$ has the following properties : $\nabla_{\mathbf{x}} f(\mathbf{x}, \mathbf{y}; \xi), \nabla_{\mathbf{y}} f(\mathbf{x}, \mathbf{y}; \xi)$ (w.r.t. $\mathbf{y}$) are Lipschitz smooth continuous with constant $L_{f_x} \geq 0, L_{f_y} \geq 0$. $\|\nabla_{\mathbf{y}} f(\mathbf{x}, \mathbf{y}; \xi)\| \leq C_{f_y}$, for some $C_{f_y} \geq 0$.
*Assumption* 3. *i)* The stochastic gradient estimate of the upper-level function satisfies: $\mathbb{E}_{\bar{\xi}}[\|\bar{\nabla} f(\mathbf{x}, \mathbf{y}; \bar{\xi}) - \mathbb{E}_{\bar{\xi}}[\bar{\nabla} f(\mathbf{x}, \mathbf{y}; \bar{\xi})]\|^2] \leq \sigma_f^2$; and *ii)* The stochastic gradient estimate of the lower-level function satisfies: $\mathbb{E}_{\zeta}[\|\nabla_{\mathbf{y}} g(\mathbf{x}, \mathbf{y}; \zeta) - \nabla_{\mathbf{y}} g(\mathbf{x}, \mathbf{y})\|^2] \leq \sigma_g^2$.

We note that Assumptions.1, 2 and 3(b) are standard in the literatures of bilevel optimization (see, e.g., Ghadimi & Wang (2018); Khanduri et al. (2021). In addition, Assumption 3(a) has been verified in (Khanduri et al., 2021).

To establish the convergence result of Prometheus, we first prove the Lipschitz-smoothness of the new gradient estimator proposed in (5), which is stated as follows:

**Lemma 1.** (Lipschitz-smoothness of the new stochastic gradient estimator in (5)). If the stochastic functions $f(\mathbf{x}, \mathbf{y}; \xi)$ and $g(\mathbf{x}, \mathbf{y}; \zeta)$ satisfy Assumptions 1–3, then we have (i) for a fixed $\mathbf{y} \in \mathbb{R}^{p_2}$, $\|\bar{\nabla} f(\mathbf{x}_1, \mathbf{y}; \bar{\xi}) - \bar{\nabla} f(\mathbf{x}_2, \mathbf{y}; \bar{\xi})\|^2 \leq L_f^2 \|\mathbf{x}_1 - \mathbf{x}_2\|^2, \forall \mathbf{x}_1, \mathbf{x}_2 \in \mathbb{R}^{p_1}$; and (ii) for a fixed $\mathbf{x} \in \mathbb{R}^{p_1}$, $\|\bar{\nabla} f(\mathbf{x}, \mathbf{y}_1; \bar{\xi}) - \bar{\nabla} f(\mathbf{x}, \mathbf{y}_2; \bar{\xi})\|^2 \leq L_f^2 \|\mathbf{y}_1 - \mathbf{y}_2\|^2, \forall \mathbf{y}_1, \mathbf{y}_2 \in \mathbb{R}^{p_2}$. In the above expressions, $L_f > 0$ is defined as: $L_f^2 := 2L_{f_x}^2 + 6C_{g_{xy}}^2 L_{f_y}^2 \left(\frac{K}{2\mu_g L_g - \mu_g^2}\right) + 6C_{f_y}^2 L_{g_{xy}}^2 \left(\frac{K}{2\mu_g L_g - \mu_g^2}\right) + 6C_{g_{xy}}^2 C_{f_y}^2 \frac{K}{L_g^2} \sum_{j=1}^{K} j^2 \left(1 - \frac{\mu_g}{L_g}\right)^{2(j-1)} \frac{1}{L_g^2} L_{g_{yy}}^2$.

*Remark* 2. We note that the Lipschitz-smoothness constant $L_f$ of Lemma 1 is smaller than that of the conventional estimator in (4), which we denote as $L_{conv}$ here, *i.e.*, $L_f \leq L_{conv}$. This also shows superiority of our new estimator. Due to space limitation, we state the definition of $L_{conv}$ in Lemma 4 in the appendix.

Next, we need the following Lipschitz-continuity properties of the approximate gradient $\bar{\nabla} f(\mathbf{x}, \mathbf{y})$, the lower level solution $\mathbf{y}*$, and the true gradient $\nabla \ell(\mathbf{x})$, which have been proved in the literature:

**Lemma 2.** (Ghadimi & Wang, 2018) Under Assumptions 1–2, we have $\|\bar{\nabla} f(\mathbf{x}, \mathbf{y}) - \nabla \ell(\mathbf{x})\| \leq L \|\mathbf{y}^*(\mathbf{x}) - \mathbf{y}\|$, $\|\mathbf{y}^*(\mathbf{x}_1) - \mathbf{y}^*(\mathbf{x}_2)\| \leq L_y \|\mathbf{x}_1 - \mathbf{x}_2\|$, $\|\nabla \ell(\mathbf{x}_1) - \nabla \ell(\mathbf{x}_2)\| \leq L_\ell \|\mathbf{x}_1 - \mathbf{x}_2\|$ for all $\mathbf{x}, \mathbf{x}_1, \mathbf{x}_2 \in \mathbb{R}^{p_1}, \mathbf{y} \in \mathbb{R}^{p_2}$, where the Lipschitz constants are defined as: $L \triangleq L_{f_x} + \frac{L_{f_y} C_{g_{xy}}}{\mu_g} + C_{f_y}\left(\frac{L_{g_{xy}}}{\mu_g} + \frac{L_{g_{yy}} C_{g_{xy}}}{\mu_g^2}\right), L_\ell \triangleq L + \frac{L C_{g_{xy}}}{\mu_g}$, and $L_y \triangleq \frac{C_{g_{xy}}}{\mu_g}$.

Lemma 2 establishes the smoothness of the implicit function in (1), which only relies on the Assumptions 1 and 2 to hold. Lastly, following the same token as in (Hong et al., 2020), we show a critical fact on the exponentially fast decay of the bias of our stochastic estimator in (5), which is stated below.

**Lemma 3** (Exponentially Decaying Bias). Under Assumptions 1–3, the stochastic gradient estimate of the upper level objective in (5) satisfies $\|\nabla f(\mathbf{x}, \mathbf{y}) - \mathbb{E}[\bar{\nabla} f(\mathbf{x}, \mathbf{y}; \bar{\xi})]\| \leq \frac{C_{g_{xy}} C_{f_y}}{\mu_g}\left(1 - \frac{\mu_g}{L_g}\right)^K$.

The assumptions and Lemmas 1-3 above lead to the main convergence result of Prometheus which is stated next.

**Theorem 1.** Under Assumptions1-3, if the step-sizes $\alpha \leq \min \left\{ \frac{(1-\lambda)m}{2\sqrt{\beta}(L_\ell+\tau)} \frac{\tau}{6+3\tau}, \frac{(1-\lambda)m}{8\sqrt{\beta}L_f^2} \frac{\tau}{6+3\tau}, \frac{\tau}{3L_\ell}, \right.$

$\left. \frac{(1-\lambda)\mu_g^2\beta^{1.5}}{23040L_y^2L^2}, \frac{8\sqrt{\beta}\tau}{12m(1-\lambda)}, \frac{20L_y^2\tau}{27(1-\lambda)\beta^{1.5}L_f^2 m}, \frac{\tau(1-\lambda)}{24mL_y^2\beta}, \frac{\tau\sqrt{\beta}(1-\lambda)}{12m}, \frac{\mu_g(1-\lambda)}{240L_y^2}\frac{\beta^{2.5}}{9L_f^2}m\frac{\tau}{6+3\tau}, (1-\lambda)\beta\frac{2m}{3}\frac{\tau}{6+3\tau} \right\},$

$\beta \leq \left\{ \frac{\sqrt{40}L_y}{3L_f}, \frac{1-\lambda}{16L_f}, (\frac{\mu_g(1-\lambda)^2}{1440L_y^2L_f^2})^2, \frac{2\mu_g}{81L_f^2} \right\}$, then the outputs of Prometheus satisfy:

$$\frac{1}{T}\sum_{t=0}^{T-1} \left[ \mathbb{E}\|\mathbf{x}_t - \mathbf{1}\otimes\bar{\mathbf{x}}_t\|^2 + \mathbb{E}\|\tilde{\mathbf{x}}_t - \mathbf{1}\otimes\bar{\mathbf{x}}_t\|^2 + \mathbb{E}\|\mathbf{y}_t - \mathbf{y}_t^*\|^2 \right] = \mathcal{O}\left(\frac{1}{T}\right).$$

*Remark* 3. It is worth noting that, compared to existing works on decentralized bilevel optimization, the major challenge in proving the convergence results in Theorem 1 stems from the *proximal operator* needed to solve the upper-level subproblem, which prevents the use of conventional descent lemma for convergence analysis (see Eq. (34) in the appendix). Also, compared to single-agent constrained bilevel optimization, one cannot provide theoretical convergence guarantee by using the direct projection method $\tilde{\mathbf{x}}_{i,t} = \arg\min_{\mathbf{x}\in\mathcal{X}}\|\mathbf{x} - (\mathbf{x}_{i,t} - \tau\mathbf{u}_{i,t})\|^2$ as in (Hong et al., 2020; Chen et al., 2022a) due to the gradient tracking procedure in the decentralized learning. Instead, we use a different proximal update rule as shown in (6). We will numerically show in Section 5 that Prometheus with the direct proximal operator can only converge to a neighborhood of a stationary point. Further, Theorem 1 implies the following sample and communication complexity results:

**Corollary 2** (Sample and Communication Complexities of Prometheus). Under the conditions of Theorem 1, to achieve an $\epsilon$-stationary solution, Prometheus requires that: i) the total number of communication rounds is $\mathcal{O}(\epsilon^{-1})$, and ii) the total number of samples is $\mathcal{O}(\sqrt{n}K\epsilon^{-1} + n)$.

### 4.4 DISCUSSION: THE BENEFIT OF VARIANCE REDUCTION IN Prometheus

Since the variance reduction in (8) in Step 3 of Prometheus requires full gradient evaluation, it is tempting to ask what is the benefit of using the variance reduction technique. In other words, could we relinquish variance reduction (VR) in Step 3 to avoid full gradient evaluation? To answer this question, consider changing Step 3 to the following basic stochastic gradient estimator without VR:

$$\mathbf{p}_i(\mathbf{x}_{i,t}, \mathbf{y}_{i,t}) = \bar{\nabla}f(\mathbf{x}_{i,t}, \mathbf{y}_{i,t}; \bar{\xi}_{i0}); \qquad \mathbf{d}_i(\mathbf{x}_{i,t}, \mathbf{y}_{i,t}) = \nabla g(\mathbf{x}_{i,t}, \mathbf{y}_{i,t}; \zeta_{i0}). \tag{10}$$

Interestingly, the following convergence result states that there always exists a *non-vanishing* constant independent of $m$, $n$, and $\alpha$ if (10) is used in Step 3 of Prometheus (i.e., a constant only dependent on problem instance and cannot be made arbitrarily small algorithmically).

**Proposition 3.** Under Assumptions1–3, with step-sizes $\alpha \leq \min\{\frac{1-\lambda}{8\beta L_f}, \frac{\tau}{3L_\ell}, \frac{(1-\lambda)m}{2\sqrt{\beta}(L_\ell+\tau)}\frac{\tau}{6+3\tau},$

$\frac{\tau\sqrt{\beta}}{6m(1-\lambda)}, \frac{\tau(1-\lambda)}{48mL_f^2\beta}, \frac{(1-\lambda)\mu_g^2\beta^{1.5}}{23040L_y^2L^2}, \mathcal{O}(T^{-\frac{1}{2}}), \frac{(1-\lambda)m}{4}\sqrt{\beta}\tau\}, \beta \leq \min\{\frac{1-\lambda}{8L_f}, \frac{(1-\lambda)^4\mu_g^2}{480^2L_y^2L_f^2}, \mathcal{O}(T^{-\frac{1}{3}})\},$ we have the following result if (10) replaces Step 3 in Prometheus,

$$\frac{1}{T}\sum_{t=0}^{T-1} \left( \mathbb{E}\|\mathbf{x}_t - \mathbf{1}\otimes\bar{\mathbf{x}}_t\|^2 + \mathbb{E}\|\tilde{\mathbf{x}}_t - \mathbf{1}\otimes\bar{\mathbf{x}}_t\|^2 \right) = \mathcal{O}\left(\frac{1}{\sqrt{T}}\right) + C'_\sigma, \tag{11}$$

where the constant $C'_\sigma$ is defined as $C'_\sigma \triangleq \frac{9(6+3\tau)}{\tau^2}\left(\left(\frac{C_{g_{xy}}C_{f_y}}{\mu_g}(1-\frac{\mu_g}{L_g})^K\right)^2 + \sigma_f^2\right) + \frac{27(1-\lambda)}{40(8+4\alpha^2)L_y^2}\frac{\beta^{1.5}}{\alpha\tau}\sigma_g^2.$

*Remark* 4. A key insight of Proposition 3 is in order. The SG-type update in (10) is similar to the SG-type update in *unconstrained* bilevel optimization in the *single-agent* setting (Ji et al., 2021). However, unlike the SG-type method in (Ji et al., 2021) that can approach zero at an $\mathcal{O}(1/\sqrt{T})$ convergence rate, the SG-type method can *only* approach a constant error $C'_\sigma$ at an $\mathcal{O}(1/\sqrt{T})$ convergence rate in the *constrained* decentralized setting. The non-vanishing constant error $C'_\sigma$ is caused by the variance $\sigma_f^2$ and $\sigma_g^2$ of the stochastic gradient . This make the benefit of using the variance reduction techniques to eliminate the $\{\sigma_f, \sigma_g\}$-variance in order to approach zero asymptotically.

## 5 NUMERICAL RESULTS

In this section, we will first conduct experiments to demonstrate the small variance of our new stochastic gradient estimator. Then, we will compare Prometheus' convergence with several baselines.

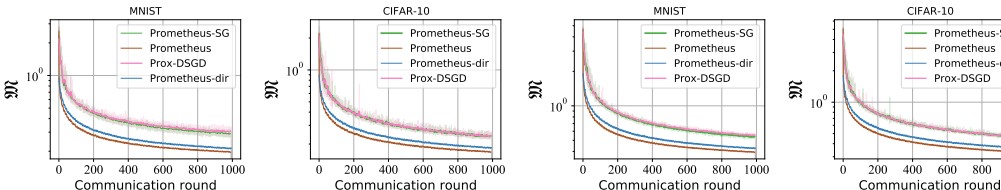

Figure 2: Five-agent network.          Figure 3: Ten-agent network.

**1) New estimator vs. conventional estimator:** Note that the major difference between the new and conventional estimators lies in how they estimate the Hessian inverse of the matrix $\mathbf{A}$. Thus, it suffices to compare the Hessian inverse approximations. The conventional estimator to estimate the $\mathbf{A}^{-1}$ can be denoted as $\tilde{\mathbf{A}}^{-1}_{conv} = K \prod_{p=1}^{k(K)}(\mathbf{I}-\mathbf{A}_s)$, while the new estimator can be denoted as $\tilde{\mathbf{A}}^{-1} = \sum_{j'=1}^{k(K)} \prod_{p=1}^{j'}(\mathbf{I}-\mathbf{A}_s)$. To see the benefits of our estimator and due to the high complexity of computing matrix inverse, here we consider a small example $\mathbf{A} = [[0.25, 0.0], [0.0, 0.25]]$, so that $\mathbf{A}_{true}^{-1} = [[4, 0], [0, 4]]$. Let $\mathbf{A}_s$ be a random matrix obtained from

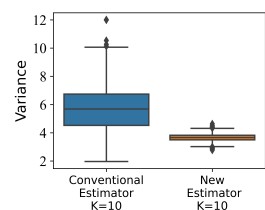

Figure 1: Hessian inverse estimator comparison.

$\mathbf{A}$ plus Gaussian noise. We use $\tilde{\mathbf{A}}^{-1}_{conv}$ and $\tilde{\mathbf{A}}^{-1}$ to estimate $\mathbf{A}^{-1}$, respectively. We run 10000 independent trials with $K = 10$ and the results are shown in Fig. 1. We can see from Fig. 1 that the new Hessian inverse estimator has a much smaller variance than the conventional one. Additional experiments on varing $K$ and different matrix $\mathbf{A}$ are relegated to our Appendix.

**2) Convergence Performance:** We verify our theoretical results of Prometheus by conducting experiments on a meta-learning problem tested on MNIST (LeCun et al., 1998) and CIFAR-10 (Krizhevsky et al., 2009) datasets. Due to space limitation, we provide additional experiments on hyper-parameter optimization in the appendix. Due to the lack of existing algorithms for solving constrained decentralized bilevel optimization problem, we compare the convergence performance of Prometheus against several stripped-down version of Prometheus:

- Prometheus *with Stochastic Gradient* (Prometheus-SG): Prometheus-SG is the SG-type algorithm discussed in Section 4.4: $\mathbf{p}_i(\mathbf{x}_{i,t}, \mathbf{y}_{i,t}) = \bar{\nabla} f(\mathbf{x}_{i,t}, \mathbf{y}_{i,t}, \bar{\xi}_{i0}); \mathbf{d}_i(\mathbf{x}_{i,t}, \mathbf{y}_{i,t}) = \nabla g(\mathbf{x}_{i,t}, \mathbf{y}_{i,t}; \zeta_{i0})$.

- Prometheus *with Direct Proximal Method* (Prometheus-dir): Instead of performing $\tilde{\mathbf{x}}_{i,t} = \arg\min_{\mathbf{x}\in\mathcal{X}}[\langle\mathbf{u}_{i,t}, \mathbf{x} - \mathbf{x}_{i,t}\rangle + \frac{\tau}{2}\|\mathbf{x} - \mathbf{x}_{i,t}\|^2 + h(\mathbf{x}_i)]$ in Prometheus, Prometheus-dir directly adds the constraints on $\mathbf{x}$: $\tilde{\mathbf{x}}_{i,t} = \arg\min_{\mathbf{x}\in\mathcal{X}}\|\mathbf{x} - (\mathbf{x}_{i,t} - \tau\mathbf{u}_{i,t})\|^2$.

- *Proximal Decentralized Stochastic Gradient Descent (Prox-DSGD):* This algorithm is motivated by the DSGD algorithm, which can be viewed as Prometheus without using gradient tracking. Specifically, we updates local gradient as $\mathbf{u}_{i,t} = \bar{\nabla} f(\mathbf{x}_{i,t}, \mathbf{y}_{i,t}; \bar{\xi}_{i0}); \mathbf{v}_{i,t} = \nabla g(\mathbf{x}_{i,t}, \mathbf{y}_{i,t}; \zeta_{i0})$.

We also note that the Prox-DSGD algorithm can be seen as a generalization of DSBO (Yang et al., 2022), SPDB (Lu et al., 2022), DSBO (Chen et al., 2022b) with the proximal operator. Prometheus - dir can also be seen as an extension of the algorithm INTERACT (Liu et al., 2020a) to handle the constrained decentralized bilevel optimization problem. We compare Prometheus with these baselines using a two-hidden-layer neural network with 20 hidden units. The consensus matrix is chosen as $\mathbf{M} = \mathbf{I} - \frac{2\mathbf{L}}{3\lambda_{\max}(\mathbf{L})}$, where $\mathbf{L}$ is the Laplacian matrix of $\mathcal{G}$ and $\lambda_{\max}(\mathbf{L})$ denotes the largest eigenvalue of $\mathbf{L}$. Due to space limitation, we relegate the detailed parameter choices of all algorithms to the appendix. In Fig. 2, we compare the performance of Prometheus, Prometheus-SG, Prometheus-dir, and Prox-DSGD on the MNIST and CIFAR-10 datasets with with a five-agent network. The network topology can be seen in Fig. 4 in Appendix D. We note that Prometheus converges much faster than than all other algorithms in terms of the total number of communication rounds. In Fig. 3, we also observe similar results when the number of tasks (and agents) is increased to 10. Our experimental results thus verify our theoretical analysis that Prometheus has the lowest communication complexity.

## 6 CONCLUSION

In this paper, we studied the constrained decentralized nonconvex-strongly-convex bilevel optimization problems. First, we proposed an algorithm called Prometheus with a new stochastic estimator. We then showed that, to achieve an $\epsilon$-stationary point, Prometheus achieves a sample complexity of $\mathcal{O}(K\sqrt{n}\epsilon^{-1} + n)$ and a communication complexity of $\mathcal{O}(\epsilon^{-1})$. Our numerical studies also showed the advantages of our proposed Prometheus and verified the theoretical results. Collectively, the results in this work contribute to the state of the art of low sample- and communication-complexity constrained decentralized bilevel learning.

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
