# OpenReview forum: "Prometheus: Endowing Low Sample and Communication Complexities to Constrained Decentralized Stochastic Bilevel Learning"
_ICLR.cc/2023/Conference — Submitted to ICLR 2023_

### Official Review · Reviewer_gfGV · 2022-10-25

**Confidence:** 3
**Clarity, Quality, Novelty And Reproducibility:** In Table 1, do you mean $m$ by $K$?
**Correctness:** 4
**Technical Novelty And Significance:** 3
**Empirical Novelty And Significance:** Not applicable
**Recommendation:** 6

**Strength And Weaknesses:**

# Strength

- I find this paper very well written. Since I am unfamiliar with the previous  literature in this topic, the clean and concise writing of this paper makes it quite easy for me to understand the intuition behind the algorithm and the key ideas in the proof.

- This setting seems to be important, and hasn't been addressed by previous works. The designed algorithm looks quite reasonable and the derived complexity aligns with my intuition for finite-sum optimization and two-timescale style algorithms.

# Weakness (and questions)

- Since the inner-loop problem is strongly convex and smooth, one could expect that (also easy to show?) the $y_i$ variable will always stay sufficiently close to $y_i^\star (x_i)$. This means the analysis of the proposed algorithm is very similar to the existing works for distributed composite smooth nonconvex optimization? Therefore, it is a bit unclear to me how much technical novelty the submission possesses in terms of either algorithmic design or techniques used in the analysis.

- This is a rather general question about the motivation of considering general topology in decentralized optimization: Can you provide a non-artificial practical problem where the network topology is not a star and the network-consensus matrix trick is indeed deployed for the sake of distributed optimization? I'd like to see some real application with supportive reference, not just some scenario that only exists in the introduction of a theory paper.

# Technical correctness

I only performed a high-level check of the proofs and they look correct to me.




**Summary Of The Paper:**

This paper studies constrained decentralized bilevel optimization where the inner-loop problem is finite-sum, smooth and strongly convex,  and the outer-loop problem is finite-sum, smooth and nonconvex. By combing (a) gradient tracking, (b) variance reduction and (c) a pseudo-gradient trick based on proximal operator, the paper designs a decentralized algorithm with both sample complexity and communication complexity matching the unconstrained setting.

**Summary Of The Review:**

Overall, I am positive with the submission and am willing to support its acceptance if my questions can be well addressed.

---

> ### Author Response · Authors · 2022-11-18
> **Response to Reviewer gfGV's comments [Part 1]**
>
> > **Your Comment 1:** Since the inner-loop problem is strongly convex and smooth, one could expect that (also easy to show?) the $y_i$ variable will always stay sufficiently close to $y_i^{\star}\left(x_i\right)$. This means the analysis of the proposed algorithm is very similar to the existing works for distributed composite smooth nonconvex optimization? Therefore, it is a bit unclear to me how much technical novelty the submission possesses in terms of either algorithmic design or techniques used in the analysis.
>
>
> **Our Response:** Thanks for your comment. First, we would like to quickly remark that the strongly-convex assumption on the lower-level problem is to ensure the uniqueness of the solution $y_i^*(x_i)$, which makes the problem well-defined. Moreover, this assumption allows a sufficient descent of the inner problem iteration so that the bilevel optimization problem is well posed and admits a tractable convergence analysis. For these reasons, this assumption has been widely adopted in most bilevel optimization papers in the literature (see, e.g., VRBO [Yang et al., 2021]; SUSTAIN [Khanduri et al., 2021]; STABLE [Chen et al., 2022a]; DSBO [Yang et al., 2022]), not just in our paper.
>
>
> We would first like to point out that in our algorithm, we do not solve $y_i$ such that it is sufficiently close to $y^\ast(x)$. Note from Algorithm 1 in the manuscript that in each round, we only update $y_i$ once, which does not ensure $y_i \approx y^\ast(x)$. In fact, it is a challenge addressed by our analysis on how to prove descent in the lower-level variable, which in turn, is coupled with the upper-level variable and still guarantees convergence in the decentralized setting with domain constraints. Moreover, just because the inner problem is strongly convex does *not* necessarily mean that the decentralized bilevel optimization problem is similar to existing works for distributed composite smooth nonconvex optimization. When the inner problem is strongly convex, it is true that one can solve for $y_i^*(x_i)$ relatively efficiently using state-of-the-art convex optimization tools. However, except for some very special inner problem structures, such a $y_i^*(x_i)$-the solution is a **numerical** solution in general and, in our problem, a crude approximation of $y_i^*(x_i)$. As a result, one cannot directly plug in the $y_i^*(x_i)$-solution to the outer-level problem as an analytical function of $x_i$. Therefore, conventional techniques for solving standard composite smooth nonconvex optimizations are *not* applicable in bilevel optimization. For example, if one wants to apply gradient-based algorithms, it is unclear how to evaluate the gradient of the outer-level problem without knowing the analytical expression of $y_i^*(x_i)$.
>
> To address the above challenges in bilevel optimization, the most popular approach in the literature so far is to leverage the *Implicit Function Theorem* to  *indirectly* evaluate a biased stochastic gradient estimate of the outer problem with respect to $x_i$. For the stochastic decentralized bilevel optimization considered in this paper, this stochastic gradient estimator can be found in Eq. (5). Note that this stochastic gradient estimator is dramatically different from those stochastic gradient estimators used in conventional composite nonconvex optimization. Moreover, due to the complex coupling between outer $x_i$-variables and inner $y_i$-variables in the stochastic bilevel gradient estimator, the convergence analysis of our algorithm is far more complex than those in standard composite nonconvex optimization, which necessitates novel proof techniques (see the appendix in this paper).

---

> > ### Author Response · Authors · 2022-11-18
> > **Response to Reviewer gfGV's comments [Part 2]**
> >
> >
> > > **Your Comment 2:** This is a rather general question about the motivation of considering general topology in decentralized optimization: Can you provide a nonartificial practical problem where the network topology is not a star and the network-consensus matrix trick is indeed deployed for the sake of distributed optimization? I'd like to see some real application with supportive reference, not just some scenario that only exists in the introduction of a theory paper which matches the decentralized problem formulation in our paper.
> >
> >
> >  **Our Response:** Thanks for your constructive comments. Decentralized optimization with the network-consensus-matrix-based approach has found quite a few real-world applications, as reported in device-to-device network optimization in [R7], decentralized training of acoustic models in [R8], and decentralized two-level transportation optimization problem in [R9], all of which are based on general graphs beyond star network topology. In these applications mentioned above, a common feature is that each node in the network can only (randomly) select a few neighboring nodes to communicate and exchange information, which matches the decentralized problem formulation in our paper and can be handled by network-consensus-matrix-based approaches.
> >
> > [R7] Asadi, Arash, Qing Wang, and Vincenzo Mancuso. "A survey on device-to-device communication in cellular networks." IEEE Communications Surveys & Tutorials 16.4 (2014): 1801-1819.
> >
> > [R8] Cui, Xiaodong, et al. "Asynchronous Decentralized Distributed Training of Acoustic Models." IEEE/ACM Transactions on Audio, Speech, and Language Processing 29 (2021): 3565-3576.
> >
> > [R9]M. Sakawa, I. Nishizaki, Y. Uemura, A decentralized two-level transportation problem in a housing material manufacturer: interactive fuzzy
> > programming approach, European Journal of Operational Research 141 (2002) 167–185
> >
> > > **Your Comments 3:** In Table 1, do you mean $m$ by $K$?
> >
> >  **Our Response:** No, $m$ and $K$ have different meanings. $K \in \mathbb{N}$ is a predefined parameter for estimating the stochastic estimator, as shown in Eqs.(5), and $m$ denotes the total number of agents in the network. In this revision, we have added a table of notations in the supplementary material and a pointer to this table in Section 4.

---

> > > ### Comment · Reviewer_gfGV · 2022-12-13
> > > **Re author response**
> > >
> > > Thank you for the detailed response. I will keep my score unchanged.

---

### Official Review · Reviewer_1rpW · 2022-10-25

**Confidence:** 3
**Correctness:** 4
**Technical Novelty And Significance:** 3
**Empirical Novelty And Significance:** 3
**Recommendation:** 6

**Clarity, Quality, Novelty And Reproducibility:**

In terms of clarity, I refer the authors to the points above. The writing is of high quality and the problem studied is novel. Providing the code or at least code samples would make the results even more reproducible.

**Strength And Weaknesses:**

Regarding strengths, to the best of my knowledge no other work has jointly combined tools from variance reduction, decentralized optimization, proximal gradients for constrained optimization and bi-level optimization in a single algorithm. All of the tools are unified in a single analysis in order to obtain an optimal sample and complexity result.

Regarding weaknesses, there are two points where additional elaboration may help make the contribution of this work clearer.

In Section 4.1.2, the authors propose a new Hessian inverse estimator that is based on the Taylor series approximation for the inverse matrix. To me it seems very similar to the Hessian inverse estimator of [1]. It also uses the same Taylor approximation formula and the same recursive update. Could the authors elaborate on the differences with [1]?

In Section 4.4, the authors highlight that variance reduction is necessary to achieve the $O(1/\epsilon)$ convergence rates in sample and communication complexity. My understanding is that this phenomenon would also appear in a stochastic optimization problem with domain constraints. This is independent of the source of the noise (sampling of $f$ or $g$) and orthogonal to the decentralized aspect. It would be appreciated if the authors would highlight which aspects of 4.4 are unique to their problem.



[1] Pang Wei Koh, Percy Liang, Understanding Black-box Predictions via Influence Functions, ICML 2017


**Summary Of The Paper:**

In this work, the authors study the problem of decentralized stochastic bi-level optimization problems with upper level constraints. The authors propose Prometheus, an algorithm that combines ideas from bi-level optimization, proximal gradient computation for constrained optimization, variance reduction for stochastic optimization as well as gradient tracking for decentralized optimization. With all of the components combined, the algorithm achieves $O(1/\epsilon)$ communication and sample complexity, a state of the art result. To demonstrate that at least some complexity is necessary, the authors showcase that even removing one component, variance reduction, can lead to solutions that do not asymptotically converge to the true solution. The theoretical results are complemented with experimental evaluations.

**Summary Of The Review:**

In summary, I am in favor of accepting this work and I am open to further increasing my score if the authors address my concerns above.

---

> ### Author Response · Authors · 2022-11-18
> **Response to Reviewer 1rpW's comments**
>
>
> > **Your Comment 1:** In Section 4.1.2, the authors propose a new Hessian inverse estimator that is based on the Taylor series approximation for the inverse matrix. To me it seems very similar to the Hessian inverse estimator of [1]. It also uses the same Taylor approximation formula and the same recursive update. Could the authors elaborate on the differences with [1]?
>
> [1] Pang Wei Koh, Percy Liang, Understanding Black-box Predictions via Influence Functions, ICML 2017
>
> **Our Response:** Thanks for your comments. At the submission time of this paper, we were unaware of Ref.  [1], which is an independent work. But we thank the reviewer for the pointer, and we will cite this paper in our revision. After comparing with Ref. [1], we agree that our Hessian inverse estimator bears some similarity to Ref. [1] in terms of using Taylor approximation. In fact, Ref. [1] is also inspired by the same previous work [Agarwal et al. (2016)] (see Page 3 in Ref. [1]). However, upon closer observations on the estimators, we note that there remain some differences between Ref. [1] and our work. The estimator in Ref. [1] is in the form of $\tilde{H} _j^{-1} = I + (I - \tilde{H}) \tilde{H} _{j-1}^{-1}$, where $\tilde{H} = \nabla _{\theta}^2 (\cdot, \hat{\theta})$, which is only designed for solving a conventional single-level minimization problem with loss function $L(\cdot)$.
>
> In contrast, our Hessian inverse estimator at each agent $i$ has the following special structure: $H_{i,k} = I + (I - (1/L_g) \nabla_{yy}^2 g(x_{i,t}, y_{i,t},\zeta_i^k)) H_{i,k-1}$, which contains a lower-level Lipschitz constant $L_g$ to tailor to the stochastic pseudo-gradient computation in Eq. (5) in *decentralized* bilevel optimization. Due to these significant structural differences, the convergence analysis under our proposed Hessian inverse estimator is far more involved.
>
>
> > **Your Comment 2:** In Section 4.4, the authors highlight that variance reduction is necessary to achieve the  $\mathcal{O}(1/\epsilon)$ convergence rates in sample and communication complexity. My understanding is that this phenomenon would also appear in a stochastic optimization problem with domain constraints. This is independent of the source of the noise (sampling of $f$ or $g$) and orthogonal to the decentralized aspect. It would be appreciated if the authors would highlight which aspects of 4.4 are unique to their problem.
>
>
> **Our Response:** Thanks for your comment. We agree with the reviewer that variance reduction (VR) can accelerate convergence rate of stochastic optimization problems in general. For example, in conventional single-level stochastic optimization, it is well-known that VR improves the convergence of the standard SGD method from $\mathcal{O}(1/\sqrt{T})$ to $\mathcal{O}(1/T)$ (implying an $\mathcal{O}(1/\epsilon)$ sample complexity). However, the effect of VR on conventional single-level stochastic optimization is relatively "minor" in the sense that, even without VR, the standard SGD method is able to converge to a stationary point with zero-norm already.
>
> However, stochastic bilevel optimization is a relatively under-explored problem, with a special structure where the upper-level problem and the lower-level problem are tightly coupled with each other. This is significantly different from standard single-level stochastic optimization. Thus, it is important to understand what impacts VR techniques could have on stochastic bilevel optimization, particularly in the decentralized setting studied in this paper. Somewhat surprisingly, as shown in Section 4.4, without using the VR technique, the SGD-style algorithm for solving stochastic decentralized bilevel optimization could only converge to an *error ball* with size $C'_{\sigma}$ at rate $\mathcal{O}(1/\sqrt{T})$. This is in stark contrast to the "zero-norm" convergence of SGD in conventional single-level stochastic optimization. In contrast, our VR-based algorithm not only recovers the "zero-norm" convergence in stochastic bilevel, but also at rate $\mathcal{O}(1/T)$. This discovery of the dramatic performance difference before and after VR in stochastic bilevel optimization is the **unique** aspect of Section 4.4.
>
>
> > **Your Comment 3:** (Clarity, Quality, Novelty And Reproducibility) In terms of clarity, I refer the authors to the points above. The writing is of high quality and the problem studied is novel. Providing the code or at least code samples would make the results even more reproducible.
>
> **Our Response:** Our source code is public on Github: https://github.com/code123share123/Prometheus.git. Thanks for pointing this out!

---

> > ### Comment · Reviewer_1rpW · 2022-11-18
> > **Why is the $1/L_g$ term necessary?**
> >
> > In their response to my first comment, the authors discuss the differences between their estimator and the one in (Koh & Liang). The main difference they pinpoint is the inclusion of an $1/L_g$ term in their gradient estimator.
> >
> > First of all, I am not sure I understand the intuition for a gradient estimator to involve the Lipschitz constants of a function. Lipschitz constants commonly appear in learning rates, iteration bounds and the analysis of algorithmic convergence. But I am not sure if I have seen an unbiased gradient estimator whose calculation depends on the (global) Lipschitz constant of the function or even a related function. Lipschitz constants are dependent on the function's global behavior whereas a function gradient describes a local behavior. Could the authors explain what is the intuition behind the $1/L_g$ term?
> >
> > Second, I checked two references that based on the author's work should also use the $1/L_g$ term (albeit not in the same way) as denoted by the authors in Equation 4. Specifically I checked (Saeed Ghadimi and Mengdi Wang) and (Liu et al.). In the first paper Equation 2.4 on page 6 of the arxiv version, does not have the $1/L_g$ term. The second paper in Equation (5) of the arxiv version also does not have the $1/L_g$ term. This is a bit problematic because the authors cite the first paper for the Lemma 2 proof and it is not clear if the gradient estimators are the same or not.
> >
> > Last but not least, the authors explain that the $1/L_g$ term is used to tailor to the *decentralized* bilevel optimization. I am not sure how the decentralized aspect of the problem affects the local lower level gradient computation.  Each agent computes their own Hessian inverse estimation and they do not need to agree on a common unbiased Hessian inverse. Based on the problem definition, there is no requirement for consensus on the lower level problem variables.
> >
> > As it is, I am not sure why the approach of (Koh & Liang), that does not have the $1/L_g$ term and thus is more intuitive, is not applicable to this problem.

---

> > > ### Author Response · Authors · 2022-11-18
> > > **Response to Reviewer 1rpW's Second Comments [Part 1]**
> > >
> > >
> > > Reviewer 1rpW
> > >
> > > > **Your Comment 1:**
> > > In their response to my first comment, the authors discuss the differences between their estimator and the one in (Koh \& Liang). The main difference they pinpoint is the inclusion of an $1 / L_g$ term in their gradient estimator.
> > > First of all, I am not sure I understand the intuition for a gradient estimator to involve the Lipschitz constants of a functions behind the $1 / L_g$ term? authors in Equation 4.
> > >
> > >
> > > **Our Response:**
> > >
> > > We thank the reviewer for the quick response. Please allow us to clarify the requirement of $1/L_g$ in our stochastic estimator. The precise reason we need the $1/L_g$ term is because the stochastic gradient estimator relies on the power series expansion of the Hessian Inverse, .i.e, $A^{-1} = \sum_{i = 0}^\infty (I - A)^i$. Please note that this power series is convergent only when the eigenvalues of $(I - A)$ are less than 1. To ensure the eigenvalues are less than 1, we have to multiply the hessian term, $\nabla^2_{yy} g(x , y; \xi)$ by $1/L_g$ as it ensures that $1/L_g \times \nabla^2_{yy} g(x , y; \xi)$ will have eigenvalue less than 1.  Otherwise, the power series will not converge. We note that this $1/L_g$ term is present in all the stochastic gradient estimators of BSA [Ref 1, Eqs.(3.62)],  INTERACT-VR [Ref 2, Eqs.(22)], TTSA [Ref 3, Page 7, Line 2], and SUSTAIN [Ref 4, Eqs.(7)].
> > >
> > >
> > >
> > >
> > > > **Your Comment 2:**  Specifically, I checked (Saeed Ghadimi and Mengdi Wang) and (Liu et al.). In the first paper, Equation $2.4$ on page 6 of the arxiv version, does not have the $1 / L_g$ term. The second paper in Equation (5) of the arxiv version also does not have the $1 / L_g$ term. This is because the authors cite the first paper for the Lemma 2 proof and it is not clear if the gradient estimators are the same or not.
> > >
> > > **Our Response:**
> > >
> > >
> > > We would like to note that both of the two papers[Ref 1, Ref 2] you've mentioned do have the term $1/L_g$.
> > >
> > > In the first paper, we would like to point out that Eqs. (2.4) in (Saeed Ghadimi and Mengdi Wang) [Ref 1] is not the stochastic estimator of $\bar{\nabla} f(x ; y)$. Specifically, Eqs. (2.4) in [Ref 1] requires computing the exact gradients and Hessian inverse which is used for a deterministic setting. In the stochastic setting [Ref 1] also utilizes a $1/L_g$ term to construct a biased stochastic estimate of the gradient, $\bar{\nabla} f({x}, {y})$, please see Eqs.(3.62) in [Ref 1]. Please see the response to the previous comment for the precise reason for the requirement of $1/L_g$ in the estimator.
> > >
> > >
> > >
> > >
> > > In the second paper, the authors use the same strategy as [Ref 1]. Please see Eqs. (22) in [Ref 2], which also has the same $1/L_g$ term.
> > >
> > > We would also like to point out that both the conventional estimator *(BSA[Ref 1, Eqs.(3.62)],  INTERACT-VR[Ref 2, Eqs.(22)], TTSA[Ref 3, Page 7, Line 2], and SUSTAIN[Ref 4, Eqs.(7)])* and our proposed estimator *(Eqs.(5) in our paper)* are stochastic estimators for constructing the stochastic gradient estimate of the deterministic estimator of Eqs. (2.4) in [Ref 1] since Eqs. (2.4) in [Ref 1]  is computationally expensive to compute, especially in a stochastic setting.
> > >
> > >
> > > Further, we would like to clarify that Lemma 2 in our paper focuses on the smoothness of the function in the general bilevel optimization problem, it is not related to the stochastic gradient estimator design we have. Specifically, in order to prove the smoothness properties as shown in our Lemma 2, one only requires Assumptions 1 and 2 to hold(for upper- and lower-level functions, respectively), however, we do not use any property of the stochastic gradient estimator. Thus, the theoretical results of Lemma 2 in our paper can be directly cited from [Ref 1].
> > >
> > > Thank you for allowing us to clarify this, we have added the above clarification in the revised manuscript.
> > >
> > >
> > >
> > >
> > >
> > >
> > >
> > >
> > > [Ref 1] Saeed Ghadimi and Mengdi Wang. Approximation methods for bilevel programming. arXiv preprint arXiv:1802.02246, 2018.
> > >
> > > [Ref 2] Zhuqing Liu, Xin Zhang, Prashant Khanduri, Songtao Lu, and Jia Liu. Interact: Achieving low sample and communication complexities in decentralized bilevel learning over networks. arXiv preprint arXiv:2207.13283, 2022.
> > >
> > > [Ref 3] Mingyi Hong, Hoi-ToWai, ZhaoranWang, and Zhuoran Yang. A two-timescale framework for bilevel optimization: Complexity analysis and application to actor-critic. arXiv preprint arXiv:2007.05170, 2020.
> > >
> > > [Ref 4] Prashant Khanduri, Siliang Zeng, Mingyi Hong, Hoi-To Wai, Zhaoran Wang, and Zhuoran Yang. A near-optimal algorithm for stochastic bilevel optimization via double-momentum. Advances in Neural Information Processing Systems, 34, 2021.
> > >
> > > [Ref 5] Xuxing Chen, Minhui Huang, and Shiqian Ma. Decentralized bilevel optimization. arXiv preprint arXiv:2206.05670, 2022b.
> > >
> > > [Ref 6] Pang Wei Koh and Percy Liang. Understanding black-box predictions via influence functions. In International conference on machine learning, pp. 1885–1894. PMLR, 2017.

---

> > > > ### Author Response · Authors · 2022-11-18
> > > > **Response to Reviewer 1rpW's Second Comments [Part 2]**
> > > >
> > > >
> > > > > **Your Comment 3:** Last but not least, the authors explain that the $1 / L_g$ term is used to tailor to the decentralized bilevel optimization. I am not sure how the decentralized problem variables.
> > > > As it is, I am not sure why the approach of (Koh \& Liang), that does not have the $1 / L_g$ term and thus is more intuitive, is not applicable to this problem.
> > > >
> > > >
> > > >
> > > > **Our Response:**
> > > > We would like to note that the $1/L_g$ term not only exists in the decentralized bilevel optimization problems like, DSBO [Ref 5, Algorithm 2, Line 7] and INTERACT-VR [Ref 2, Eqs. (22)], but also exists in the single-agent bilevel optimization problems, e.g., BSA [Ref 1, Eqs.(3.62)], TTSA [Ref 3, Page 7, Line 2], SUSTAIN [Ref 4, Eqs.(7)]. Our proposed stochastic estimator can not only be used in decentralized bilevel learning but also sheds light on single-agent bilevel learning especially for solving non-smooth regularizers in the upper-level problems.
> > > >
> > > >
> > > > We would like to point out that (Koh \& Liang)[Ref 6] does not have $1/L_g$ term because the authors assume w.l.o.g. that the Hessian $\nabla^2_{yy} g(x,y) \preceq 1$ (please see the footnote in page 4), which implies that the authors implicitly assume $L_g = 1$. Note that we do not make any such assumption on our loss functions and thereby we need to scale the stochastic estimators of $\nabla^2_{yy} g(x,y)$ with $1/L_g$. We would also like to point out that although the approach of (Koh \& Liang)[Ref 6] to estimate the Hessian inverse shares some similarities to ours, [Ref 6] is only designed for solving a conventional single-level minimization problem with loss function $L(\cdot)$.
> > > >
> > > >
> > > >
> > > >
> > > > [Ref 1] Saeed Ghadimi and Mengdi Wang. Approximation methods for bilevel programming. arXiv preprint arXiv:1802.02246, 2018.
> > > >
> > > > [Ref 2] Zhuqing Liu, Xin Zhang, Prashant Khanduri, Songtao Lu, and Jia Liu. Interact: Achieving low sample and communication complexities in decentralized bilevel learning over networks. arXiv preprint arXiv:2207.13283, 2022.
> > > >
> > > > [Ref 3] Mingyi Hong, Hoi-ToWai, ZhaoranWang, and Zhuoran Yang. A two-timescale framework for bilevel optimization: Complexity analysis and application to actor-critic. arXiv preprint arXiv:2007.05170, 2020.
> > > >
> > > > [Ref 4] Prashant Khanduri, Siliang Zeng, Mingyi Hong, Hoi-To Wai, Zhaoran Wang, and Zhuoran Yang. A near-optimal algorithm for stochastic bilevel optimization via double-momentum. Advances in Neural Information Processing Systems, 34, 2021.
> > > >
> > > > [Ref 5] Xuxing Chen, Minhui Huang, and Shiqian Ma. Decentralized bilevel optimization. arXiv preprint arXiv:2206.05670, 2022b.
> > > >
> > > > [Ref 6] Pang Wei Koh and Percy Liang. Understanding black-box predictions via influence functions. In International conference on machine learning, pp. 1885–1894. PMLR, 2017.

---

> > > > > ### Comment · Reviewer_1rpW · 2022-11-18
> > > > > **Very thorough and clear response**
> > > > >
> > > > > I would like to thank the authors for the clear and though response. Based on my understanding, the difference between how the authors and (Koh & Liang) handle the normalization of $g$ is the following: (Koh & Liang) assumes that $g$ is normalized in advance, hence no $1/L_g$ term, whereas the authors normalize during the algorithm execution by dividing with $1/L_g$ for every reference to $g$.
> > > > >
> > > > > Even if (Koh & Liang)'s approach was designed for influence function applications and the authors' version is designed for bi-level optimization, at the end of the day both estimate the same quantities with the same formula (up to who is responsible for $g$'s normalization).
> > > > >
> > > > > The paper currently suggests that the estimator of (Koh & Liang) is merely related to theirs. I encourage the authors to be more precise about the differences in their presentation.

---

> > > > > > ### Author Response · Authors · 2022-11-22
> > > > > > **Response to Reviewer 1rpW's Comments**
> > > > > >
> > > > > > Thanks for your positive and constructive comments! We have incorporated your suggestions and comments in the revised version of this paper; see details on page 5.

---

> > ### Comment · Reviewer_1rpW · 2022-11-18
> > **Decreasing learning rates and convergence**
> >
> > The authors suggest that in classical stochastic non-variance reduced approaches, noise variance affects the convergence rates but does not affect the ability of SGD to converge to stationary points. At least for the [standard SGD convergence analysis](https://www.cs.ubc.ca/~schmidtm/Courses/540-W19/L11.pdf), if one were to choose a constant learning rate one would not get convergence to a stationary point but to an error ball that is a function of the noise variance. It is only through non-constant learning rates that one gets convergence to stationary points.
> >
> > It seems to me that in 4.4 the analysis for the non-variance reduced algorithm is also done with a constant learning rate. So it seems that the convergence to an error ball is not surprising since this phenomenon is also present in classical stochastic non-variance reduced optimization.
> >
> > Decreasing learning rates reduce the effective noise variance in the gradient estimation. Intuitively, instead of taking multiple gradient samples and averaging to reduce variance, decreasing learning rates allow us to take repeated samples from nearby points. I understand that advanced variance reduction methods have better sample efficiency, but it is unclear why the decreasing learning rate technique would fail completely in achieving zero norm convergence in this setting.
> >
> > If the authors could elaborate more on why the decreasing learning rate technique does not work here, I would be grateful. As it is, I am not sure what part of 4.4 is unique to the setting studied by the authors.

---

> > > ### Author Response · Authors · 2022-11-22
> > > **Response to Reviewer 1rpW's 3rd Comments [Part 1]**
> > >
> > > > **Your Comment 1:** The authors suggest that in classical stochastic non-variance reduced approaches, noise variance affects the convergence rates but does not affect the ability of SGD to converge to stationary points. At least for the standard SGD convergence analysis, if one were to choose a constant learning rate one would not get convergence to a stationary point but to an error ball that is a function of the noise variance. It is only through non-constant learning rates that one gets convergence to stationary points.
> > >
> > > **Our Response:** Thanks for your comments. First, we would like to point out that your comment, "The standard SGD might reach stationary points if we utilize the diminishing learning rates." is only true for asymptotic analysis where the iteration $T$ goes to infinity. In the finite-time convergence analysis where the number of iterations $T$ is given, it is possible to converge arbitrarily close to a stationary point using a fixed step-size. Note also that our paper also considers finite-time convergence analysis, where $T$ is given.
> > >
> > > For example, in SGD, with the knowledge of the total number of iterations $T$, a constant learning rate of the order $\mathcal{O}(1/\sqrt{T})$ can guarantee an $\mathcal{O}(1/\sqrt{T})$ convergence rate for conventional non-bilevel stochastic minimization problems, see Corollary 2.2 in [R1]. Similar conclusions can also be found in decentralized non-convex non-bilevel minimization problems. Please see Corollary 2 in [R2] and Theorem 1 in [R3], where the convergence rate results do not have an error ball even though the learning rate is a constant. Therefore, it is not necessary to use non-constant learning rates to ensure convergence to stationary points in finite-time convergence analysis.
> > >
> > > [R1] Ghadimi, Saeed, and Guanghui Lan. "Stochastic first-and zeroth-order methods for nonconvex stochastic programming." SIAM Journal on Optimization 23.4 (2013): 2341-2368
> > >
> > > [R2] Lian, Xiangru, et al. "Can decentralized algorithms outperform centralized algorithms? a case study for decentralized parallel stochastic gradient descent." Advances in Neural Information Processing Systems 30 (2017).
> > >
> > > [R3] Yu, Hao, Rong Jin, and Sen Yang. "On the linear speedup analysis of communication efficient momentum SGD for distributed non-convex optimization." International Conference on Machine Learning. PMLR, 2019

---

> > > > ### Author Response · Authors · 2022-11-22
> > > > **Response to Reviewer 1rpW's 3rd Comments [Part 2]**
> > > >
> > > > > **Your Comment 2:**  It seems to me that in 4.4 the analysis for the non-variance reduced algorithm is also done with a constant learning rate. So it seems that the convergence to an error ball is not surprising since this phenomenon is also present in classical stochastic non-variance reduced optimization. Decreasing learning rates reduce the effective noise variance in the gradient estimation. Intuitively, instead of taking multiple gradient samples and averaging to reduce variance, decreasing learning rates allow us to take repeated samples from nearby points. I understand that advanced variance reduction methods have better sample efficiency, **but it is unclear why the decreasing learning rate technique would fail completely in achieving zero norm convergence in this setting. If the authors could elaborate more on why the decreasing  learning rate technique does not work here, I would be grateful. As it is, I am not sure what part of 4.4 is unique to the setting studied by the authors.**
> > > >
> > > >
> > > >
> > > > **Our Response:** Thank you for your comments. The main reason for "the failing of decreasing learning rate technique" is due to the fact that our decentralized bilevel problem contains i) a non-smooth term $h(\mathbf{x})$ in the loss function, and ii) the domain constraints. To handle these complications, we utilize the proximal operators shown in Eq. (6) in our paper, which is restated below:
> > > > $$\widetilde{{\mathbf{x}}} _{i,t}= \tilde{\mathbf{x}}(\mathbf{x} _{i, t})=\arg \min _{\mathbf{x} \in \mathcal{X}}\left[\left\langle\mathbf{u} _{i, t}, \mathbf{x}-\mathbf{x} _{i, t}\right\rangle+\frac{\tau}{2}\left||\mathbf{x}-\mathbf{x} _{i, t}\right||^2+h(\mathbf{x})\right],$$
> > > > where $\tau$ is a fixed constant that can be interpreted as the weight of the proximal regularizer. Here, it is **important** to note that the constant $\tau$ is independent of $T$. To see this, as we mentioned in the paper, Eq.(6) is inspired by the SONATA method [R4], where the function $\left\langle\mathbf{u} _{i, t}, \mathbf{x}-\mathbf{x} _{i, t}\right\rangle+\frac{\tau}{2}\left||\mathbf{x}-\mathbf{x} _{i, t}\right||^2$ is the first-order (Taylor) approximation that plays the role of a surrogate function of the loss function $\ell({\mathbf{x}})$. Therefore, $\tau$ is the constant that is associated with the higher-order residual terms in the first-order approximation, which is independent of the number of iterations $T$.
> > > >
> > > > Now, consider the error term $C' _\sigma$. It can be seen from Eq. (86) in our paper that, given the choice of the step-sizes $\alpha$ and $\beta$, $C _\sigma'$ can be written as:
> > > >
> > > >
> > > > $$C _{\sigma}'= \frac{9(6+3\tau)}{\tau^2}((\frac{C _{g _{x y}} C _{f _{y}}}{\mu _{g}} (1-\frac{\mu _{g}}{L _{g}} )^{K})^2+\sigma _f^2)+  \frac{27 (1-\lambda)}{40(8+4\alpha^2)L _y^2} \frac{\beta^{1.5}}{\alpha\tau} \sigma _g^2.$$
> > > >
> > > > It can be seen from the above equation that the step-size choices for $\alpha$ and $\beta$ **will only affect the second term $\frac{27 (1-\lambda)}{40(8+4\alpha^2)L_y^2} \frac{\beta^{1.5}}{\alpha\tau} \sigma_g^2$ in $C'_\sigma$**. As a result, $C_{\sigma}'$ will **always contain a non-vanishing constant error term** $\frac{9(6+3\tau)}{\tau^2}((\frac{C_{g_{x y}} C_{f_{y}}}{\mu_{g}} (1-\frac{\mu_{g}}{L_{g}} )^{K})^2+\sigma_f^2)$, **no matter how one picks diminishing step-sizes for $\alpha$ and $\beta$.**
> > > >
> > > > [R4] Scutari, Gesualdo, and Ying Sun. "Distributed nonconvex constrained optimization over time-varying digraphs." Mathematical Programming 176.1 (2019): 497-544.

---

### Official Review · Reviewer_Rqm7 · 2022-11-04

**Confidence:** 3
**Correctness:** 3
**Technical Novelty And Significance:** 3
**Empirical Novelty And Significance:** Not applicable
**Recommendation:** 5

**Clarity, Quality, Novelty And Reproducibility:**

- (Quality and Clarity) The paper is reasonably well-written, though I believe a table of notation would be immensely helpful. There is so much notation in the paper and it is very easy to get lost.

- (Reproducibility) While the main contribution of the paper is theoretical, the code used to run experiments is not provided. Therefore, the paper's experiments are not reproducible. This reduces the value of the paper for future work that would seek to build on it.

- (Novelty) While the paper relies on SARAH/SPIDER-style estimators for variance reduction, it introduces a new estimator for the stochastic gradient of the bilevel problem that is interesting in its own right. The algorithm Prometheus as a whole isn't very novel outside of this, and as the authors state can be seen as an extension of INTERACT-VR [1] to the proximal case.


[1] Zhuqing Liu, Xin Zhang, Prashant Khanduri, Songtao Lu, and Jia Liu. INTERACT: Achieving Low Sample and Communication Complexities in Decentralized Bilevel Learning over Networks. MobiHoc 2022

**Strength And Weaknesses:**

- (Strength) The introduced algorithm, Prometheus, achieves a sample complexity of $\mathcal{O} (\sqrt{n} K \epsilon^{-1} + n)$. This is currently the best-known sample complexity for finite-sum stochastic bilevel optimization problems. The same can be said for the communication complexity $\mathcal{O} (\epsilon^{-1})$.

- (Strength) The proposed algorithm can handle complex constraints through the use of the proximal operator, enabling more applications than previous algorithms.

- (Strength) The authors introduce a new recursive estimator for the stochastic gradient of the bilevel problem (see eq. 5), and this new estimator seems to show some benefits in practice over the conventional estimator used by previous works.

- (Weakness) The authors do not give many examples of the constraints their algorithm can solve, and their relevance to practice. The motivating examples all do not really make use of the new proximal formulation.

- (Weakness) The benefits of the new recursive estimator are only illustrated in one experiment, and it is difficult to see the benefit since Figure 1 only tests out this estimator on a single matrix with fixed norm. Can you plot how the estimator performs for varying norm of ||A^-1||? How about with increasing variance? There is too little information to figure out how it performs empirically.

- (Weakness) It is not clear why the ordinary inverse Hessian estimator does not work for this problem, given that it works just fine for INTERACT-VR and achieves the same rate for the unconstrained setting.

- (Weakness) The discussion about the necessity of VR is a little misleading. The authors introduce a variant of Prometheus without variance reduction, and then give an upper bound for this variant that shows it does worse than the variance-reduced version. However, this is not a correct way to show the necessity of some technique-- at best, it provides some evidence for this and nothing more. The correct way would be to give a lower bound, but no such lower bound is given here.

- (Weakness) The authors do not really explain why the proximal setting is significantly more difficult than the projected setting. In ordinary, non-bilevel, stochastic optimization, the extension from the projected case to the proximal case is very simple and straightforward. What is the source of difficulty here?

**Summary Of The Paper:**

This paper introduces a new algorithm, Prometheus, for the composite stochastic bilevel optimization problem in the decentralized setting. In particular, the paper considers the setting where we have a distributed optimization problem on $m$ agents, each of which has a loss function of the form $l(x) + h(x)$ where $l(x)$ has bilevel, finite-sum structure and $h$ is a constraint function with an easy-to-compute proximal operator. The agents are connected to each other via a decentralized network (i.e. there is no central node to exchange iterates with, and therefore the agents have to solve a consensus problem as well during the optimization process). The bilevel optimization problem is nonconvex-strongly-convex (the outer function is nonconvex and smooth, the inner function is strongly convex), this problem has been solved before in the decentralized setting, but only with simple projection constraints (e.g. as in [1]). The algorithm introduced here, Prometheus, solves this problem in the setting where $h$ can be an arbitrary constraint but whose proximal operator is easily computed. Prometheus achieves the same communication complexity and local sample complexity as INTERACT-VR from [1], but can do so under more difficult constraints.

[1] Zhuqing Liu, Xin Zhang, Prashant Khanduri, Songtao Lu, and Jia Liu. INTERACT: Achieving Low Sample and Communication Complexities in Decentralized Bilevel Learning over Networks. MobiHoc 2022

**Summary Of The Review:**

I think this is a paper which is quite borderline, but for which the weaknesses outweigh the strengths. My biggest concern with this paper is that I am not sure where the difficulty is in the new proximal setting. I cannot really see it in the proof, and I don't think it is highlighted enough. Moreover, the experimental evaluation is not enough to see the benefits of some of the proposed techniques (like the new stochastic gradient estimator). The discussion on the necessity of variance reduction is somewhat flawed (for the reason outlined above), and as such I'd prefer to see that section rewritten as well.

-----------------------

Post-rebuttal summary:
I thank the reviewers for their long, detailed responses. The examples given are satisfying to me, and I think there is some technical difficulty that the authors have overcome here. However, after discussion with other reviewers, I am not convinced the technical novelty here may not be enough, especially given that the Hessian estimator used is not novel. As such, I can not recommend acceptance.

---

> ### Author Response · Authors · 2022-11-18
> **Response to Reviewer Rqm7's Comments [Part 1]**
>
> Thank you for your constructive feedback and comments. Our point-to-point responses are as follows:
>
> > **Your Comment 1:**  The authors do not give many examples of the constraints their algorithm can solve, and their relevance to practice. The motivating examples all do not really make use of the new proximal formulation.
>
> **Our Response:**
>
> Thanks for your comments. As we mentioned in the paper, the constrained decentralized bilevel optimization problem has attracted increasing attention due to its foundational role in many emerging multi-agent learning paradigms, particularly with **regularization constraints**. Thus, our algorithm can be adopted in many real-world applications. In our paper, we have provided some simple applications in Sections 1 and 3.2. Here, we would like to further elaborate and provide more concrete examples on the real-world applications with constraints that can be modeled and solved by our constrained decentralized bilevel problem formulation:
>
>
> 1. **Sparsity-regularized meta-learning [R1]:*** Sparsity-regularized optimization problems are widely seen in the machine learning community, which is one of the promising tools for high-dimensional machine learning with guaranteed statistical efficiency and robustness to overfitting. In particular, [R1] has theoretically and numerically justified that sparsity would enhance the robustness of optimization-based meta-learning. Specifically, the problem in [R1] is formulated as $\min _\theta \mathcal{R}(\theta),\text { s.t. }||\theta_l
> ||_0 \leq k_l, l \in[L]$, where $L$ is the total number of network layers.
>
> This problem can be reformulated as $\min_{x\in\mathcal{X}} \ell(x)+h(x)$ following the standard Tikhonov regularization approach and matching the formulation in our paper. For the constraint in [R1], $||\theta_l||_0$ indicates the number of non-zero entries in the parameters of $l$-th layer $\theta_l$ that is required to be no larger than a user-specified sparsity level $k_l$. In the sparsity-regularized optimization literature, it is widely accepted to replace $||\theta_l ||_0$ by $||\theta_l ||_1$ since $\ell_1$-norm also promotes sparsity.
>
>
>
>
>
>
> 2. ***Rank-constrained decentralized matrix completion for recommender systems [R2]:*** In matrix completion problems, the basic idea is to use the sparsity and low rank of the matrices to impute values where entries are missing. Also, the rank-constrained algorithm sequentially computes low-rank SVDs that could help the algorithms fit in memory. As shown in Eqs. (7.4) in [R2], the problem is formulated in the following constrained bilevel form:
>
> $\min_{u \in X_{a d}} J(u, \mu)=\mathcal{J}(u)+\mathcal{R}(u,\mu)  $
>
> $\text { s.t. } \min_{\mu\in\mathcal{M}_{a d}}\phi(\mu),$
>
> where $\phi(\cdot)$ denotes the upper-level objective function and is specified by the user, $\mathcal{J}$ is the true objective and $\mathcal{R}$ is the regularization with $\mu$ denoting the hyper-parameter.
> $X_{ad}$ and $\mathcal{M}_{ad}$ represent rank-constraints in $u$ or $\mu$.
>
>
>
>
>
> 3. ***Hyper-parameter LASSO[R3]:*** The Hyper-parameter LASSO problem, such as Leave-one-out (LOO)-validated Lasso and Group Lasso can be reformulated as a constrained bilevel problem, where the constraint comes from the penalty parameter. For example, the hyper-parameter LASSO problem in [R3] is formulated as follows:
>
> $\min _w  \ell(w)+\lambda^{\top} \Omega(w).$
>
> $\text { s.t. } \lambda^*(\lambda, \mathcal{B}):=\arg \min _{\lambda }\ell^{\mathrm{VAL}}\left(w^*\left(\lambda, \mathcal{B}\right)\right)).$
>
>
>
> [R1] Tian, Hongduan, et al. "Meta-learning with network pruning." European Conference on Computer Vision. Springer, Cham, 2020
>
>
> [R2]Panagoda, Mahendra Harshin. Convergence Analysis and Bilevel Optimization Algorithms for Matrix Completion Problems. Diss. George Mason University, 2021.
>
> [R3]Lopez-Ramos, Luis M., and Baltasar Beferull-Lozano. "Online Hyperparameter Search Interleaved with Proximal Parameter Updates." 2020 28th European Signal Processing Conference (EUSIPCO). IEEE, 2021.

---

> > ### Author Response · Authors · 2022-11-18
> > **Response to Reviewer Rqm7's comments [Part 2] }**
> >
> >
> > > **Your Comment 2:**  The benefits of the new recursive estimator are only illustrated in one experiment, and it is difficult to see the benefit since Figure 1 only tests out this estimator on a single matrix with fixed norm. Can you plot how the estimator performs for varying norm of $||A^{-1}||$? How about with increasing variance? There is too little information to figure out how it performs empirically.
> > > The experimental evaluation is not enough to see the benefits of some of the proposed techniques (like the new stochastic gradient estimator).
> >
> > **Our Response:** Thanks for your suggestion. Due to the space limitation at the submission time, we had put additional experiments for our new estimator in our Appendix (see Page 33 in the appendix). We conducted experiments on different $A$ matrices by varying the $K$-value to test our new estimator. From these experiments, we can see that our new estimator always has a smaller variance than the conventional estimator.
> >
> >
> > > **Your Comment 3:** It is not clear why the ordinary inverse Hessian estimator does not work for this problem, given that it works just fine for INTERACT-VR and achieves the same rate for the unconstrained setting.
> >
> > **Our Response:** We thank the reviewer for raising this concern. We would like to clarify that we did not claim that the ordinary Hessian inverse estimator does not work for this problem. It is true that both Hessian estimators achieve the same $\mathcal{O}(1/T)$ convergence rate. Instead of improving the convergence rate order in terms of $T$-dependence, our goal in this paper is to improve the hidden constant in the Big-O by proposing a new Hessian inverse estimator with a smaller variance compared to that of the conventional estimator. Such an improvement in the hidden Big-O constant will subsequently lead to other benefits in convergence. Specifically, in Section 4.3, we theoretically show that our proposed new estimator has a smaller Lipschitz constant $L_f$ than the conventional Lipschitz constant $L_{conv}$.  In Section 5 and Appendix, we also numerically demonstrate the smaller variance of our new estimator compared to that of the conventional estimator.
> >
> >
> >
> >
> >
> >
> >
> >
> > > **Your Comment 4:** The discussion about the necessity of VR is a little misleading. The authors introduce a variant of Prometheus without variance reduction, and then give an upper bound for this variant that shows it does worse than the variance-reduced version. However, this is not a correct way to show the necessity of some technique-- at best, it provides some evidence for this and nothing more. The correct way would be to give a lower bound, but no such lower bound is given here.
> > The discussion on the necessity of variance reduction is somewhat flawed (for the reason outlined above), and as such I'd prefer to see that section rewritten as well.
> >
> > **Our Response:** Thanks for your constructive comments. We agree that the wording "necessity" is imprecise, which has caused some confusion. Here, the meaning of the word "necessity" is not in the strict sense of logic. Rather, what we wanted to express is the benefit of using VR. As we pointed out in Section 4.4, the benefit of using VR is reflected in the dramatic improvement on the convergence rate. Specifically, before using VR, we have a $\mathcal{O}(1/\sqrt{T})$ convergence rate approaching a constant error $C'_\sigma$. In contrast, after using VR, we achieve convergence to the exact stationary point at rate $\mathcal{O}(1/T)$. To avoid this confusion, we have replaced the word  "necessity" by "benefit" in the revised version.
> >
> > Meanwhile, we also agree that one cannot draw a comparative conclusion regarding the convergence performances of two algorithms by simply comparing their convergence rate upper bounds. To reach a definite conclusion, convergence rate lower bounds (i.e., fundamental convergence rate limits) of the considered algorithms are also needed. However, deriving tight lower bounds for stochastic optimization algorithms is often highly challenging, which often warrants an independent study by itself. Thus, most of the papers in this field still resort to comparisons among convergence rate upper bounds (see the summary tables in [VRDBO(Gao et al., 2022), DSBO (Chen et al., 2022b), SPDB (Lu et al., 2022), SUSTAIN (Khanduri et al., 2021), VRBO (Yang et al., 2021),STABLE (Chen et al., 2022a)]. In this paper, we still follow this convention in the literature in our performance discussions in Section 4.4. We would also like to point out that establishing the lower bounds on the performance of stochastic bilevel optimization problems is an open problem and is an interesting future direction. Again, we thank the reviewer for these constructive comments!

---

> > > ### Author Response · Authors · 2022-11-18
> > > **Response to Reviewer Rqm7's comments [Part 3]**
> > >
> > > > **Your Comment 5:** The authors do not really explain why the proximal setting is significantly more difficult than the projected setting. In ordinary, non-bilevel, stochastic optimization, the extension from the projected case to the proximal case is very simple and straightforward. What is the source of difficulty here?
> > > > My biggest concern with this paper is that I am not sure where the difficulty is in the new proximal setting. I cannot really see it in the proof, and I don't think it is highlighted enough.
> > >
> > > **Our Response:** Thanks for your constructive comments. To see the difficulty in the new proximal setting, it is helpful to first revisit the problem formulation restated as follows:
> > > i
> > > $\min_{ x_i  \in X } \frac{1}{m}\sum_{i=1}^m  [\ell({x_i})+h(x_i)] \triangleq \frac{1}{mn}\sum_{i=1}^m\sum_{j=1}^n [ f ( {x}_i, {y}_i^{*} ( x_i ; {\xi} _{ij} ))+ h(x_i)]$
> > >
> > > $\text{s.t.} ~~ {{y}}_i^{*}({{x}_i}) =\arg\min _{y}  g({{x}_i}, {{y}}_i ) \triangleq \frac{1}{n}  \sum _{j=1}^n   g({{x}_i}, {{y}}_i ; {\zeta} _{ij} ) ,  \forall i; \quad {x} _i = {x} _{i'},   ~~ \text{if }~~ (i, i') \in \mathcal{L}.$
> > >
> > >
> > > The algorithm design for solving the problem above faces a number of challenges: (i) the objective function $\mathbf{x}$ is nonconvex; (ii) the objective function is nonsmooth; (iii) the constraint set on $\mathbf{x}$-variables; (iv) the decentralized bilevel problem structure.
> > >
> > > Although challenges (i)-(iii) have been somewhat addressed in ordinary single-level stochastic optimization (as you pointed out), it is the combination of all these challenges, particularly the *coupling* between proximal operation (for addressing challenges (ii) and (iii)) and the decentralized bi-level structure, that renders the theoretical analysis of algorithm design extremely challenging in proving our Prometheus algorithm to be both sample- and communication-efficient. To overcome the challenges (i)-(iv) mentioned above, we propose the first provably convergent algorithmic framework called Prometheus for solving the general class of constrained decentralized bilevel optimization problems.
> > >
> > >
> > >
> > >
> > > In particular, regarding your question on the specific technical challenges in the proximal setting in decentralized bilevel optimization, we would like to emphasize that the major difficulty stems from theoretically establishing the consensus of the upper-level variables under the decentralized topology. The challenge arises when we perform projections directly on ${x}$ to ensure the feasibility of local ${x}_i$-variables:
> > >
> > >
> > > ${{x} } _{i, t+1}= \mathcal{P} _{ \mathcal{X} }  ({{{x}} } _{i,t}-\alpha {u} _{i,t} ) = \arg\min _{ \tilde{x} \in \mathcal{X} }  || \tilde{x} - ({{{x}} } _{i,t}-\alpha {u} _{i,t} )  ||^2 .$
> > >
> > >
> > >
> > > To address this challenge, rather than performing a projection directly as above, we propose to use an auxiliary variable $\tilde{\mathbf{x}}_{i,t }$ to tackle the problem indirectly. As a result, the iterative update of local $\mathbf{x}_i$-variables is based on the following new successive convex approximation (SCA) technique:
> > >
> > >
> > > $\tilde{x}_{i} ({{x}} _{i,t}) ={\arg\min} _{{{x}} \in \mathcal{X}} [ \langle {u} _{i,t}, {x}- {x} _{i,t}\rangle +\frac{\tau}{2} \| {{x}}- {{x}} _{i,t}\|^2+h({x}) ];$
> > >
> > >
> > > ${x} _{i,t+1} =\sum _{i' \in \mathcal{N} _{i}} [M] _{i i'} \mathbf{x} _{i',t} + \alpha(	\tilde{{x}}_i( {x} _{i,t}) - {x} _{i,t}).$
> > >
> > > The used auxiliary proximal operator $\tilde{{x}} _{i,t}$ and the resultant local update $\alpha( \tilde{{x}} _i ({{x}} _{i,t}) - {x} _{i,t})$ in the consensus step (see the second equation above) play an important role in helping us tackle the non-smooth objective challenge. This SCA technique is dramatically different from the conventional algorithm design in ordinary single-level stochastic optimization. Without this new SCA technique, it will be difficult, if not entirely impossible, to achieve convergence guarantees. Moreover, the use of the above new SCA technique also necessitates many proof techniques that are quite different from the proofs in ordinary single-level stochastic optimization (see the proof details of our Lemma 5 and Lemma 7 in the Appendix). To clarify the major challenges, we have added the above discussions in the revised version.

---

> > > > ### Author Response · Authors · 2022-11-18
> > > > **Response to Reviewer Rqm7's comments [Part 4]**
> > > >
> > > > > **Your Comments 6:**  (Quality and Clarity) The paper is reasonably well-written, though I believe a table of notation would be immensely helpful. There is so much notation in the paper, and it is very easy to get lost.
> > > >
> > > > **Our Response:** Thanks for your suggestion. We do agree that adding a table of notations would be very helpful. At the time of submission, we didn't add a notation table due to space limitations. In this revision, we have added a table of notations in the supplementary material and a pointer to this table in Section 4.
> > > >
> > > > Also, we want to clarify in here that having complex notations is the nature of bilevel optimization problems even in the centralized (single-agent) setting. The notation complexity of bilevel optimization is further exacerbated in decentralized multi-agent settings. In fact, we use typical notations as those in the related work ([R6, R7, R8]).
> > > >
> > > >
> > > > [R4] Hongchang Gao, Bin Gu, and My T Thai. Stochastic bilevel distributed optimization over a network. arXiv preprint arXiv:2206.15025, 2022.
> > > >
> > > > [R5] Shuoguang Yang, Xuezhou Zhang, and Mengdi Wang. Decentralized gossip-based stochastic bilevel optimization over communication networks. arXiv preprint arXiv:2206.10870, 2022.
> > > >
> > > > [R6] Zhuqing Liu, Xin Zhang, Prashant Khanduri, Songtao Lu, and Jia Liu. Interact: Achieving low sample and communication complexities in decentralized bilevel learning over networks. arXiv preprint arXiv:2207.13283, 2022.
> > > >
> > > > > **Your Comments 7:** (Reproducibility) While the main contribution of the paper is theoretical, the code used to run experiments is not provided. Therefore, the paper's experiments are not reproducible. This reduces the value of the paper for future work that would seek to build on it.
> > > >
> > > > **Our Response:** Our source code is public on Github: https://github.com/code123share123/Prometheus.git.
> > > > We overlooked this since we didn't notice any place in the submission page regarding the source code. Thanks for pointing this out!
> > > >
> > > > > **Your Comments 8:** (Novelty) While the paper relies on SARAH/SPIDER-style estimators for variance reduction, it introduces a new estimator for the stochastic gradient of the bilevel problem that is interesting in its own right. The algorithm Prometheus as a whole isn't very novel outside of this, and as the authors state can be seen as an extension of INTERACT-VR [1] to the proximal case.
> > > > [1] Zhuqing Liu, Xin Zhang, Prashant Khanduri, Songtao Lu, and Jia Liu. INTERACT: Achieving Low Sample and Communication Complexities in Decentralized Bilevel Learning over Networks. MobiHoc 2022]
> > > >
> > > >
> > > > **Our Response:** Thanks for your comments. We would like to emphasize again the novelty of our proposed Prometheus algorithm. As we mentioned in our response to your Comment 5, although our VR technique is inspired by the SARAH/SPIDER-style estimators for variance reduction, the algorithm design and analysis of Prometheus are far more challenging and complex than those in INTERACT-VR and by no means a straightforward extension of INTERACT-VR. Specifically, here we highlight two major key differences compared to INTERACT-VR:
> > > >
> > > > * INTERACT-VR cannot handle non-smooth objectives considered in our work. To tackle this challenge, we propose a special proximal operator $\tilde{{x}} _{i}({x} _{i, t})$ as mentioned in our response to your Comment 5 above, which is dramatically different from those used in ordinary single-level stochastic optimization. We show that this special-structured proximal operator not only makes our Prometheus algorithm numerically efficient but also renders the convergence analysis of Prometheus theoretically tractable. As shown in our experiments, without the term $\tilde{{x}} _{i}({x} _{i, t})$, the convergence of the direct proximal version of the INTERACT-VR algorithm (called Prometheus-dir) may not even converge to the stationary point.
> > > >
> > > > * The Prometheus algorithm integrates a new stochastic gradient estimator, as opposed to the conventional stochastic gradient estimator adopted in INTERACT-VR. Notably, in Section 4.3, we theoretically show that our proposed new stochastic gradient estimator enjoys a smaller Lipschitz constant $L_f$ than the conventional Lipschitz constant $L_{conv}$. These theoretical results are new in the literature. In Section 5 and Appendix, we also numerically demonstrate the smaller variance of our new stochastic gradient estimator over the conventional one.

---

### Decision · Program_Chairs · 2023-01-20

**Decision:**

Reject

**Justification For Why Not Higher Score:**

This paper reads incremental comparing to Liu et al. 2022, the additional technical novelty may not be sufficient for ICLR.

**Justification For Why Not Lower Score:**

N/A

**Metareview: Summary, Strengths And Weaknesses:**

This paper considers the problem of constrained decentralized stochastic bilevel optimization, and proposed an algorithm that achieve the same rate as Liu et al. 2022, but can additionally address the constrained and non-smooth setting. Compared to Liu et al. 2022, the algorithm proposed in this paper further use techniques of the proximal steps and different Hessian estimator.

This is a borderline paper has both pros and cons. During the AC-reviewer meeting, reviewers agree that the results are new and solid. However, they found that proximal steps relatively standard in optimization to dealing with constraints and non-smoothness. Even though reviewers agree that the proximal extension may not be trivial in the setting of bilevel optimization, they are not convinced why analyzing the proximal steps are very challenging in this setting. The Hessian estimator used in this paper is almost the same as Koh and Liang 2017. Although authors claim difference in two "key aspects" in the revised version, both differences look minor. We believe this paper would improve significantly if it can properly highlight the true challenges of this paper compared to Liu et al. 2022, and justify why the new results are novel and significant (instead of just a relatively expected extension from prior works). Given the current form we recommend rejection.

**Summary Of Ac-Reviewer Meeting:**

Strength:
1. (40%) It's a strict generalization of previous results. The results are solid and new.

Weakness:
1. (60%) The results are a bit incremental comparing to Liu et al 2022: all additional techniques used seem standard. It's not very clear what are the major technical challenges.